# Learning Homophilic Incentives in Sequential Social Dilemmas

## Abstract

Promoting cooperation among self-interested agents is a long-standing and interdisciplinary problem, but receives less attention in multi-agent reinforcement learning (MARL). Game-theoretical studies reveal that altruistic incentives are critical to the emergence of cooperation but their analyses are limited to non-sequential social dilemmas. Recent works using deep MARL also show that learning to incentivize other agents has the potential to promote cooperation in more realistic sequential social dilemmas (SSDs). However, we find that, with incentivizing mechanisms, the team cooperation level does not converge and regularly oscillates between cooperation and defection during learning. We show that a second-order social dilemma resulting from the incentive mechanisms is the main reason for such fragile cooperation. We analyze the dynamics of second-order social dilemmas and find that a typical tendency of humans, called homophily, provides a promising solution. We propose a novel learning framework to encourage homophilic incentives and show that it achieves stable cooperation in both SSDs of public goods and tragedy of the commons.

## 1 Introduction

Deep multi-agent reinforcement learning (MARL) has achieved prominent progress (Sunehag et al., 2018; Rashid et al., 2018; Baker et al., 2020; Wang et al., 2021). While much effort is devoted to fully cooperative settings, decision-making individuals in many real-world situations may be self-interested, such as autonomous vehicles (Bonnefon et al., 2016), taxpayers (Rothstein, 2001), and governments dealing with climate change (Capstick, 2013). A unique phenomenon among self-interested agents is social dilemma, in which individually rational behavior results in a situation where everyone suffers in the long run. Historically, the problem of how cooperation emerges in social dilemmas has long perplexed scientists from various fields, such as evolutionary biology, animal behavioristics (Packer et al., 1991), and neuroscience (Rilling et al., 2002), and is listed as one of the 125 most compelling science questions for the 21st century (Kennedy, 2005; Pennisi, 2005).

Cooperation emergence in non-sequential social dilemmas (SDs) have been extensively studied (Boyd, 1989; Rand & Nowak, 2011). However, these methods only consider the problems that can be cast as a matrix game (Hughes et al., 2018), and typically assume the existence of atomic actions for cooperative and defective strategies. Thus their applicability to real-world, temporally extended SDs is limited, where cooperation and defection are policies that need to be learned. With the ability to learn policies in complex tasks, deep MARL shows promise to fill the gap and study how cooperation emerges in these more realistic sequential social dilemmas (SSDs) (Leibo et al., 2017).

Some previous works have studied SSDs from the perspective of MARL (Jaques et al., 2019; Hughes et al., 2018). Among them, Yang et al. (2020) and Lupu & Precup (2020) extend (altruistic) incentives to temporally extended cases. Referring to individuals paying costs to punish or reward others even though there is no immediate gain by these actions for themselves (Ostrom et al., 1992; Güth, 1995; Fehr & Gächter, 2002; Boyd et al., 2003; Mussweiler & Ockenfels, 2013), altruistic incentives have been proven to be highly related to cooperation emergence in non-sequential SDs (Fowler, 2005; Akçay & Roughgarden, 2011; Fong & Surti, 2009). However, altruistic incentives have not been fully studied in SSDs. The aforementioned deep methods do not effectively scale up and are limited in very small settings. With further investigation, we find that these methods do not converge to stable cooperation, where the cooperation level regularly *oscillates* between cooperation and defection.

In this paper, we first analyze the underlying reasons behind the phenomenon of such unstable cooperation. We find that, although incentives make cooperation more likely to emerge, they also introduce a second-order social dilemma (2nd-SDs) problem into the system – if someone else would pay costs to punish or reward others, why should I bother to do so (Dreber et al., 2008)? The consequence is that more and more second-order free-riders would exploit those agents who make an effort to incentivize others, resulting in a degraded incentivizing mechanism and thus collapsed cooperation. 2nd-SDs have been well studied in non-sequential social dilemmas (Fowler, 2005; Greenwood, 2016). However, solving 2nd-SDs in sequential settings remains largely untouched in the literature and posts new challenges – the identification of cooperation and defection for incentivizing is non-trivial in SSDs and cooperation is more vulnerable to 2nd-SDs in SSDs than in SDs. Thus SSDs require a different computational approach to promote cooperation. Specifically, in SSDs, agents need to learn temporally extended strategies which are much more complex than simple cooperation and defection action in SDs and can be dynamically mixed with cooperation and defection behaviors, especially during the learning. The agents that have successfully learned cooperative incentivizing policies can be exploited by other agents with temporally learned mixing strategies, and the exploited cooperators may abandon cooperative strategies, making it less likely for other agents to learn to cooperate. This issue does not exist in SDs but will severely hinder cooperation emergence in SSDs.

To solve this problem, we first illustrate the dynamics of and formally analyze the impetus for the second-order social dilemma on a fully-featured motivating example. Based on these analyses, we propose a novel learning mechanism by encouraging agents with similar environmental behaviors to have similar incentivizing behaviors. Our method is inspired by a concept called *homophily*, a particularly common tendency for human individuals to associate or bound with similar others, as in the proverb *birds of a feather flock together*. In the literature, homophily has been studied in non-sequential social dilemmas, where agents interact with others of similar predefined types (Fu et al., 2012; Fletcher & Doebeli, 2009; Ramazi et al., 2016). In non-sequential cases, agents can choose their partners to interact with (Rand et al., 2011; Aksoy, 2015) or there is a matching system that auto-matches similar agents (Bergstrom, 2003; Bilancini et al., 2016), but it is not clear how to extend these insights to SSDs. Our work solves this problem and enables a group of self-interested agents to cooperate stably when they are *continually learning* and acting in a *shared environment*.

It is worth noting that multi-person social dilemmas are classified into two broad categories: public goods and the tragedy of the commons (HARDİN, 1968; Kollock, 1998; Ledyard, 1994; Hardin, 2009). We evaluate our method on both of these sequential social dilemmas and show that the agent population learns stable cooperative behaviors with great efficiency and stability. Visualization of the evolution process of cooperation shows that homophily effectively enables stable cooperation by preventing agents that conduct altruistic incentives from being exploited by second-order free-riders.

## 2    PRELIMINARIES AND RELATED WORKS

Social dilemmas (SDs) are among the most important settings used to study the emergence of cooperation. In a social dilemma, there exists a tension between the individual and collective rationality, in which individually rational behavior results in a situation where everyone suffers in the long run. Although studies on SDs have contributed significantly to the research of cooperation emergence for decades (Axelrod & Hamilton, 1987; Peysakhovich & Lerer, 2017; Anastassacos et al., 2020), they focus on fixed policies, where the cooperative and defective strategies are actions to be chosen rather than policies to be learned. To be more realistic as in real-world situations, in this paper, we consider **sequential social dilemmas** (SSDs, (Leibo et al., 2017; Blanco et al., 2014)).

An SSD can be modelled as a partially-observable general-sum Markov game (Littman, 1994) $\mathcal{M} = \langle I, S, \{A_i\}, P, O, \{\Omega_i\}, \{R_i\}, n, \gamma \rangle$, where $I$ is a finite set of $n$ agents and $\gamma$ is a discount factor. At each time step, agent $i$ draws a partial observation $o_i \in \Omega_i$ of the state $s \in S$ according to the observation function $O(s, i)$. Based on the observation, agent $i$ selects an action $a_i \in A_i$, which together form a joint action $\boldsymbol{a}$, leading to a next state $s'$ according to the stochastic transition function $P(s'|s, \boldsymbol{a})$ and individual rewards $r_i = R_i(s, \boldsymbol{a})$ for each agent. In SSDs, instead of taking atomic cooperation or defection actions, agents must learn cooperation or defection strategies consisting of potentially long sequences of environmental actions. The goal of each agent is to maximize the local expected return: $Q_i(s, \boldsymbol{a}) = \mathbb{E}_{s_{0:\infty}, \boldsymbol{a}_{0:\infty}} [\sum_{t=0}^{\infty} \gamma^t R_i(s_t, \boldsymbol{a}_t) | s_0 = s, \boldsymbol{a}_0 = \boldsymbol{a}]$.

**Altruistic incentive**, including altruistically rewarding and punishing others, is known as one of the solutions for social dilemmas (Kollock, 1998). However, it introduces the problem of **second-order social dilemma** (2nd-SD), which arises from each individual's inclination to free ride on a mechanism that is designed to solve the first-order social dilemma. To solve 2nd-SDs, previous works either introduce additional mechanisms, such as extra punishing mechanisms (Fowler, 2005; Greenwood, 2016), reputation mechanisms (Panchanathan & Boyd, 2004), or change the game settings, such as enabling the amount of public goods to grow exponentially with the number of contributors (Ye et al., 2016), considering corruption and power asymmetries (Úbeda & Duéñez-Guzmán, 2011). These works analyze the dynamics of predefined and fixed mechanisms and focus on non-sequential games.

We propose to solve 2nd-SDs in general cases using the tendency of **homophily**, which states that agents tend to have similar incentivizing behaviors if their environmental behaviors are similar. In the literature, homophily, frequently termed as assortative matching (Bergstrom, 2003), is often referred to the tendency for agents to interact with others of similar types (Fu et al., 2012; Fletcher & Doebeli, 2009; Ramazi et al., 2016). These works assume that a game (such as Prisoner's Dilemma and Public Goods Dilemma) involves only a group of several agents from a large population. Based on the way they form the group, these works can be divided into two broad categories. (1) Agents can choose their partners to interact with (Rand et al., 2011; Aksoy, 2015). However, in SSDs, agents do not have such an option and their partners are determined at initialization. They have to continually learn and act in a shared environment instead of a separate environment instance for each group. (2) There is a matching system that can auto-match agents with similar predefined types to join the game (Bergstrom, 2003; Bilancini et al., 2016). However, in SSDs, agents are assumed to have determined partners and thus the matching process changes the game settings. Moreover, the existence of such a matching system is an unrealistic assumption because the policy types (such as cooperation and defection) in SSDs cannot be predefined since temporally extended strategies are much more complex and can be dynamically mixed with cooperation and defection, especially during learning. Therefore, previous work studying homophily can not be applied to SSDs. Moreover, Wang et al. (2018) also find that assortative matching of *environmental* behaviors cannot promote cooperation in SSDs. In contrast, our work encourages homophily on the level of *incentivizing* behaviors, which we find outperforming other algorithms on learning cooperation in SSDs. Furthermore, we discuss how homophily relates to other possible solutions in Appendix B.9.

Recent works study some other mechanisms which may encourage cooperation in SSDs. Hughes et al. (2018) study the effects of inequality aversion, which bypasses the problem of second-order social dilemmas because the punishment does not incur costs to any agents other than the punished ones. Jaques et al. (2019) find that encouraging mutual influence among agents can promote collective behaviors. We empirically compare with these methods in Sec. 5.2.

In the following sections, we will first provide a motivating example to show the influence of SDs, 2nd-SDs, and how homophily can solve 2nd-SDs. Based on these analyses, we introduce how to encourage homophily in temporally extended cases in Sec. 4.

## 3  MOTIVATING EXAMPLE: FRAGILE COOPERATION

In this section, we provide a detailed motivating example to explain first-order social dilemmas (1st-SDs) and second-order social dilemmas (2nd-SDs) in the context of RL, and demonstrate that homophily can promote cooperation by alleviating the 2nd-SDs. For clarity, in this section, we use one-step games to illustrate our idea, which inspires our extension to sequential settings in Sec. 4.

We adopt a classic problem setting of public goods dilemma (Hauert et al., 2007; 2002a; Semmann et al., 2003; Hauert et al., 2002b). A population of $n$ agents has an opportunity to create a public good, from which all can benefit, regardless of whether they have contributed to the good. Specifically, there are three strategies (atomic actions). Contributors ($C$) pay a cost $c$ to increase the size of public good by $b$. Defectors ($D$) do not contribute. The public good is uniformly distributed among cooperators and defectors. Agents can also choose to neither contribute to nor benefit from the public good, but receive a fixed reward $\sigma$, as Nonparticipants ($N$). For clarity, we conduct analyses using an illustrative example with $n$=10, $b$=3, $c$=1, $\sigma$=1. We also provide a sensitivity analysis regarding parameters in Appendix B.3 to consolidate that our conclusion generally holds.

**A: Cooperation is fragile in 1st-SDs.** We first showcase the influence of 1st-SDs. The Schelling diagram (Schelling, 1973; Perolat et al., 2017) (Fig. 1(a)) proves the existence of SDs in this case. To visualize the learning dynamics of MARL algorithms under 1st-SDs, we learn independent policies for agents using REINFORCE (Williams, 1992). In Fig. 1(b), we plot the change of the proportion of cooperative agents in the population under 3 random seeds. We observe that the cooperation level oscillates during learning. For explanation, we visualize the dynamics of population constituent in a ternary plot (Fig. 1(c)). Each point $X$ inside the equilateral triangle represents a distribution of population members $(p_C, p_D, p_N)$, where $p_C + p_D + p_N = 1$, $p_C, p_D, p_N$ are represented by the distances from point $X$ to the edges $ND$, $CN$, and $CD$, respectively. Trajectories in Fig. 1(c) correspond to the curves with the same color in Fig. 1(b). We can observe that all the trajectories rotate counterclockwise in the vicinity of vertex $N$ regardless of the starting position.

Formally, we calculate the closed-form gradients of agents' true value functions and visualize the *gradient field* in Fig. 1(d). It can be observed that cooperation is not a stable solution, which can be easily taken over by defectors, and then by nonparticipants. This phenomenon is the result of 1st-SDs, where cooperation is exploited by defection, eventually leading the system to a very ineffective state. We refer readers to Appendix E for the detailed proof.

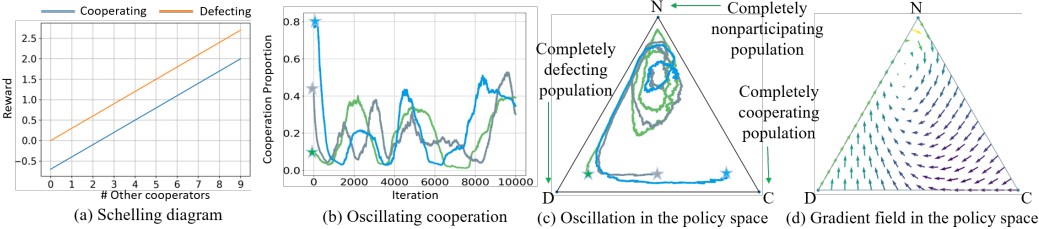

(a) Schelling diagram     (b) Oscillating cooperation     (c) Oscillation in the policy space     (d) Gradient field in the policy space

Figure 1: First-order social dilemmas.

**B: Unexploitable altruistic incentives make cooperation possible**. To introduce altruistic incentives into the system, we add Punishers ($P$) as the fourth type of strategy. The same as contributors, a punisher contributes to and benefits from the public good, and importantly, it also pays a cost $k$ to incur a punishment $p$ on defectors. This punishment is altruistic because it reduces its own immediate reward but benefits the team in the long run. To show the effect of altruistic incentives, we need to guarantee that no agent can exploit altruistic incentives by being a pure contributor that does not pay cost to punish defectors. Therefore, we first consider the setting where punishers also pay a cost $\alpha k$ to incur a punishment $\alpha p$ on the pure contributors for not punishing defectors (Fowler, 2005). We denote this punishing type by $PA$, which represents a kind of *unexploitable altruistic incentives*.

For simplicity, we illustrate our analyses with $p$=2, $k$=0.35, $\alpha$=1 (sensitivity analysis is deferred to Appendix B.3). Each horizontal plane in Fig. 2(a) shows the Schelling diagram under the corresponding number of first-order defectors. Similar to Fig. 1(c), we visualize the dynamics of independent learners under this situation in Fig. 2(b) using a quaternary plot (refer to Appendix A.2 for more details). We see that although two trajectories are trapped in the $C$-$D$-$N$ plane and similar to those in Fig. 1(c), one of the three trajectories finds the stable cooperation solution $PA$.

We plot the closed-form gradients in Fig. 3(a), where the green (blue) region indicates that a population initialized there would converge to cooperative (non-cooperative) solution. This figure proves that introducing unexploitable altruistic incentives creates a "safe region" near $PA$, and the populations initialized there converge to cooperative solutions.

**C: Exploitable altruistic incentives suffer from 2nd-SDs, which again lead to fragile cooperation.** Now we restrict the punishments incurred by punishers, and they can only pay a cost $k$ to incur a punishment $p$ on defectors. Now the punishers can be exploited by pure cooperators who do not pay for but benefit from punishments. We call this type of punishers the *exploitable punishers*. On each horizontal plane in Fig. 2(c), we show the Schelling diagram with different numbers of first-order defectors, which reveals the existence of second-order social dilemmas.

Formally, assume that the probability of agent $i$ taking three actions are $\theta^{i,C}, \theta^{i,D}, \theta^{i,P}$, respectively. It can be derived (Appendix E) that the gradient of agent $i$'s value function w.r.t. the probability of second-order cooperative action $\theta^{i,P}$ minus the the gradient w.r.t the probability of second-order defective action $\theta^{i,C}$ is $-k \sum_{j \neq i} \theta^{j,D}$, where $k > 0$ is the punishment cost. The gradient is negative

for any $\theta^{j,D}$, which indicates that second-order cooperators would be taken over by second-order defectors. This is the second-order social dilemma. The result is that only strategies $C, D, N$ exist, after which the system degrades due to the 1st-SD as discussed before. We plot the closed-form gradients in Fig. 2(d), from which we observe that the "safe region" disappears, and, for any initialization, the population ends with non-cooperation.

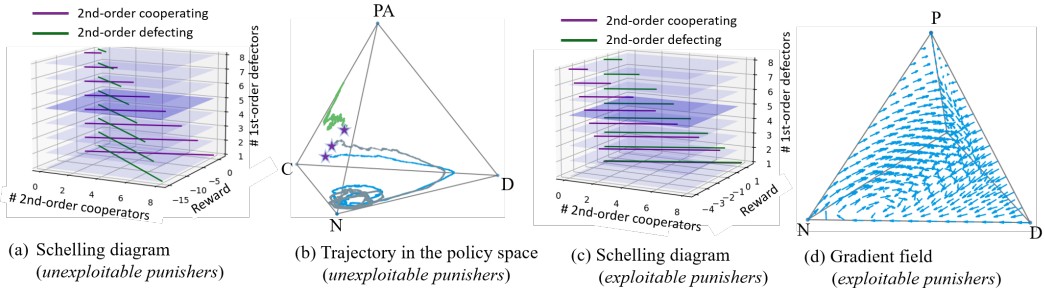

(a) Schelling diagram (*unexploitable punishers*)  (b) Trajectory in the policy space (*unexploitable punishers*)  (c) Schelling diagram (*exploitable punishers*)  (d) Gradient field (*exploitable punishers*)

Figure 2: Unexploitable and exploitable altruistic incentives.

**Takeaways** We conclude that only unexploitable incentives can make cooperation a stable solution. It becomes particularly problematic in temporally extended cases, where the incentivizing policies need to be learned. It is typical that some agents would learn altruistic incentives earlier than others. However, as analysed before, these altruistic agents will be exploited by other agents, leading to degraded altruistic behaviors. Further, with collapsed altruistic incentivizing mechanisms, the population falls back to a 1st-SD, making cooperation much less likely to emerge.

**D: Homophily solves second-order social dilemmas.** To show the effect of homophily, based on the settings in part C, we further encourage agents with similar acting behaviors to have similar incentivizing behaviors. Since only contributors and punishers have the same acting behavior of contributing to the public good, we encourage their incentives to be the same by converting the minority of P and C to the majority with a probability of 0.2.

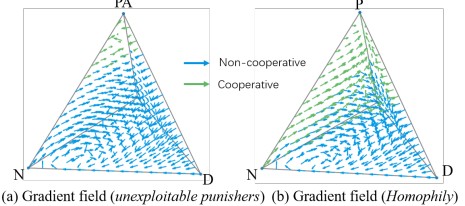

(a) Gradient field (*unexploitable punishers*) (b) Gradient field (*Homophily*)

Figure 3: Homophily solves second-order social dilemmas.

We plot the closed-form gradient in Fig. 3(b). We find that the "safe region" reappears. The gradient w.r.t. $\theta^{i,P}$ minus the gradient w.r.t $\theta^{i,C}$ is $-k\sum_{j\neq i}\theta^{j,D} + 2\lambda\texttt{sign}(\sum_{j\neq i}(\theta^{j,P} - \theta^{j,C}))\min(\theta^{i,C}, \theta^{j,P})$, which is positive when close to point P. Interestingly, the "safe region" is larger than that in Fig. 3(a). In this way, we conclude that homophily helps solve 2nd-SDs.

By this motivating example, we show that homophilic incentives encourage cooperation. However, this study focuses on one-step games and predefines the mechanism of punishers. The question is how to encourage homophily in realistic settings without depending on any predefined mechanisms.

## 4 METHODS

As discussed in the previous section, 2nd-SDs disturb the learning process of incentivizing actions – agents who learn altruistic incentives earlier tend to be exploited by other agents. Consequently, the altruistic incentivizing behaviors are typically taken over and the population would fall back to 1st-SDs, resulting in a hopeless loop.

In this section, we discuss how to introduce homophily into sequential social dilemmas so that the loop can be broken and cooperation is possible to emerge and stabilize. We propose to encourage homophily by encouraging agents with similar acting (environmental) behaviors to have similar incentivizing behaviors. We now describe our learning framework shown in Fig. 4.

### 4.1 INCENTIVIZING AND ENVIRONMENTAL BEHAVIOR LEARNING

To enable incentivizing behaviors, we add incentivizing actions to the action space. We use $a_{i\rightarrow j}$ to denote the incentivizing action from $i$ to $j$. The action $a_{i\rightarrow j}$ induces an inter-agent reward $\eta^e r_{i\rightarrow j}$

to agent $j$. Here, $\eta^e > 0$ is a scaling factor. In this paper, we consider three types of incentivizing actions with a positive, negative, and zero $r_{i \to j}$, respectively. Since we consider altruistic incentives, the action $a_{i \to j}$ itself costs $\eta^c |r_{i \to j}|$, where $\eta^c > 0$ is also a scaling factor.

Each agent learns two Q functions, for selecting environmental actions and incentivizing actions, respectively. At each step, agents first simultaneously select environmental actions $a_i$ according to $Q_{\theta_i}^{i,\text{env}}(\tau_i, a_i)$, which is based on local action-observation history and is parameterized by $\theta_i$. Then, conditioned on environmental actions of other agents, $\boldsymbol{a}_{-i}$, each agent $i$ decides its incentivizing actions $\boldsymbol{a}_{i \to -i}$ according to $Q_{\phi_i}^{i,\text{inc}}((\tau_i, \boldsymbol{a}_{-i}), \boldsymbol{a}_{i \to -i})$ parameterized by $\phi_i$.

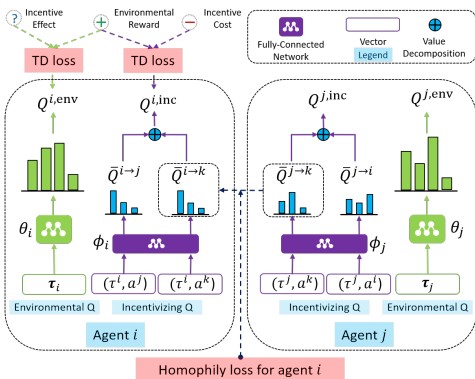

One question is what rewards should be considered when training $Q_{\phi_i}^{i,\text{inc}}$. Intuitively, incentivizing actions are expected to positively influence the return given by the environment. Therefore, we include environment rewards $r_i$. We also consider the costs of incentivizing actions to prevent agents from excessively giving incentives. Moreover, we ignore the

Figure 4: Homophily learning framework.

rewards received from other agents, which can effectively prevent agents from learning trivial and detrimental policies, such as keeping exchanging positive incentives regardless of the observations.

Another question of learning $Q_{\phi_i}^{i,\text{inc}}$ is that it requires $3^{n-1}$ outputs using a conventional deep Q-network and most output heads would remain unchanged for long stretches of time. To solve this problem, agent $i$ can learn $n - 1$ incentivizing Q functions $\bar{Q}_{\phi_i}^{i,\text{inc}}$, each of which corresponds to one agent $j \neq i$. However, this alternative arises a new question because the environment rewards are considered when training $Q_{\phi_i}^{i,\text{inc}}$ but they do not present an explicit decomposition over agents. To solve this problem, we propose to estimate $Q_{\phi_i}^{i,\text{inc}}$ as a summation:

$$Q_{\phi_i}^{i,\text{inc}}((\tau_i, \boldsymbol{a}_{-i}), \boldsymbol{a}_{i \to -i}) = \sum_{j \neq i} \bar{Q}_{\phi_i}^{i,\text{inc}}((\tau_i, a_j), a_{i \to j}). \tag{1}$$

Here, parameters of $\bar{Q}_{\phi_i}^{i,\text{inc}}$ are shared to accelerate training. This formulation is similar to VDN (Sunehag et al., 2018), but we sum incentivizing Q's of a single agent, rather than Q's of different agents.

With these formulations, we train each agent's incentivizing Q by minimizing the following TD loss:

$$\mathcal{L}_i^{\text{inc}}(\phi_i) = \mathbb{E}_{\mathcal{D}}\Big[\big(r_i - \eta^c \sum_{j \neq i} |r_{i \to j}| + \gamma^{\text{inc}} \max_{\boldsymbol{a}_{i \to -i}'} Q_{\phi_i^-}^{i,\text{inc}}((\tau_i', \boldsymbol{a}_{-i}'), \boldsymbol{a}_{i \to -i}') - Q_{\phi_i}^{i,\text{inc}}((\tau_i, \boldsymbol{a}_{-i}), \boldsymbol{a}_{i \to -i})\big)^2\Big],$$
$$\tag{2}$$

where $\gamma^{\text{inc}}$ is the discount factor, $\phi_i^-$ is parameters of a *target network* that are periodically copied from $\phi_i$, and the expectation is estimated by uniform samples from a replay buffer $\mathcal{D}$.

Environmental Q function is trained with rewards from the environment and the incentives received from other agents, and we minimize the following TD loss for learning $Q^{i,\text{env}}$:

$$\mathcal{L}_i^{\text{env}}(\theta_i) = \mathbb{E}_{\mathcal{D}}\Big[\big(y_i^{\text{env}} - Q_{\theta_i}^{i,\text{env}}(\tau_i, a_i)\big)^2\Big]. \tag{3}$$

Here, the expectation is estimated with uniform samples from the replay buffer $\mathcal{D}$, $y_i^{\text{env}} = r_i + \eta^e \sum_{j \neq i} r_{j \to i} + \gamma^{\text{env}} \max_{a_i'} Q_{\theta_i^-}^{i,\text{env}}(\tau_i', a_i')$ is the target for environmental Q-learning. $\theta_i^-$ is parameters of a target network that are periodically copied from $\theta_i$.

## 4.2 HOMOPHILY

Directly learning incentivizing policies can be difficult due to second-order social dilemmas as we discussed in Sec. 3. To solve this problem and inspired by the stateless case in Sec. 3 Part D, we encourage agents to be homophilic, *i.e.*, agents with similar environmental behaviors should have

Figure 5: Comparison of our method against baselines and ablations.

similar incentive behaviors, which can be expressed as a loss to be *minimized*:

$$\mathcal{L}_i^{\text{homo}}(\phi_i) = \mathbb{E}_{\mathcal{D}}\Big[-\sum_{j \neq i} \mathcal{S}^{\text{env}}(i,j)\mathcal{S}^{\text{inc}}(i,j;\phi_i)\Big], \tag{4}$$

where $\mathcal{S}^{\text{env}}$ is 1 or 0, indicting the environmental behaviors are similar or not. $\mathcal{S}^{\text{inc}}$ measures the similarity between incentivizing behaviors of two agents.

The first question is how to define $\mathcal{S}^{\text{inc}}(i,j;\phi_i)$. The idea is to measure the similarity between two agents' incentive behaviors by comparing their incentive actions to each of other agents. For each agent $i$, we measure its similarity to agent $j$ as:

$$\mathcal{S}^{\text{inc}}(i,j;\phi_i) = -\sum_{k \notin \{i,j\}} \mathcal{CE}\Big[P_{a_{j \to k}} \,\big\|\, \sigma(\bar{Q}_{\phi_i}^{i,\text{inc}}((\tau_i, a_k), \cdot))\Big], \tag{5}$$

where $\sigma(\cdot)$ is the softmax function, $P_{a_{j \to k}}$ is a categorical distribution over agent $j$'s incentive actions, with a probability of 1 for the action that agent $j$ takes for agent $k$ ($a_{j \to k}$). The cross entropy $\mathcal{CE}[\cdot\|\cdot]$ measures the distance between these two distributions.

As for $\mathcal{S}^{\text{env}}$, if we parameterize and learn it with $\mathcal{S}^{\text{inc}}$ using $\mathcal{L}_i^{\text{homo}}$, we may get trivial solutions. For example, $\mathcal{S}^{\text{env}}$ may be a constant 1, in which case $\mathcal{L}_i^{\text{homo}}$ is minimized given the same $\mathcal{S}^{\text{inc}}$. To avoid such solutions, we propose to use non-parametric $\mathcal{S}^{\text{env}}$. Details are discussed in Appendix C.

With these components, the loss for agent $i$ to learn environmental and incentivizing behaviors is:

$$\mathcal{L}_i(\theta_i, \phi_i) = \mathcal{L}_i^{\text{env}}(\theta_i) + \lambda^{\text{inc}}\mathcal{L}_i^{\text{inc}}(\phi_i) + \lambda^{\text{homo}}\mathcal{L}_i^{\text{homo}}(\phi_i), \tag{6}$$

where $\lambda^{\text{inc}}$ and $\lambda^{\text{homo}}$ are scaling factors. Agents learn their policies independently.

## 5 EXPERIMENTS

Our experiments aim to answer the following questions: (1) Can homophily promote the emergence of cooperation? (Sec. 5.2) (2) What is the contribution of each component in the proposed learning framework? (Sec. 5.3) (3) How does cooperation emerge and evolve under homophilic incentives? (Sec. 5.4) (4) How does homophily affect incentive behaviors? (Sec. 5.5)

### 5.1 EXPERIMENTAL SETUP

We test our method in SSDs. There are two broad categories of multi-person social dilemmas (Kollock, 1998): *Public goods dilemmas* and *Tragedy of commons dilemmas*. In this paper, we consider sequential versions of these two dilemmas, Cleanup and Harvest (Leibo et al., 2017). In our learning framework, each agent has an environmental and an incentivizing Q function. The Q network architecture is the same for all agents but they do not share parameters. For other details of environments and our method, we refer readers to Appendix C.

### 5.2 PERFORMANCE

To test whether homophilic learning can promote the emergence of cooperation, we test our method on Cleanup and Harvest with different numbers of agents and compare against various baselines: LIO (Yang et al., 2020), Inequity Aversion (Hughes et al., 2018), Social Influence (Jaques et al., 2019), and Selfish Actor-Critic. The details of baselines can be found in Appendix D.

We test all methods with 5 random seeds and show the mean value as well as 95% confidence intervals in Fig. 5. It can be observed that our method helps agents learn cooperation with efficiency and

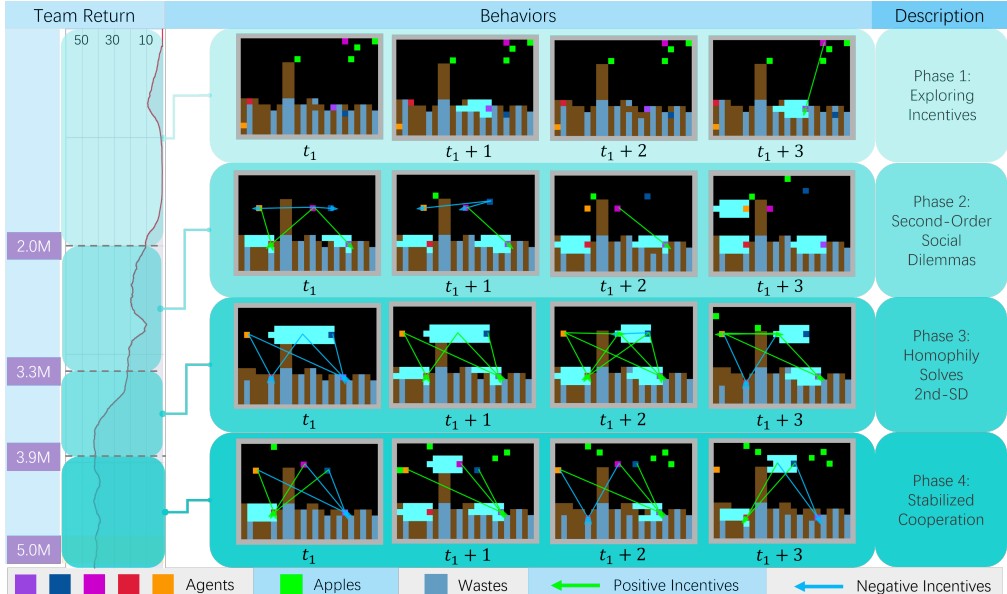

Figure 6: Evolution of cooperation in `Cleanup`. Left: Sum of environmental rewards received by all agents. Middle: Environmental and incentivizing behaviors at four different stages of learning. Right: Corresponding descriptions.

stability. In comparison, baseline algorithms either cannot learn any cooperation strategies or the cooperation level oscillates. *Inequity aversion* oscillates on `Cleanup` (n=5), which can be proved by its large confidence intervals, and cannot learn to cooperate on `Harvest` (n=10). *Social influence* proves that cooperation can be achieved by encouraging agents to influence each other, but it requires many samples (typically 100M as reported in Jaques et al. (2019)) to learn cooperative strategies. In contrast, our method needs around 5M samples to learn stable cooperative strategies. LIO can learn cooperation on `Cleanup` (n=3), but is less effective in dilemmas with more agents.

## 5.3 ABLATION STUDY

There are three contributions that characterize our method. (1) First and the most important, the homophily learning objective. (2) Discrete incentive actions and factored incentive Q-functions for each agent. (3) Excluding received incentives when training incentive Q-functions. In this section, we design the following ablations to test the contribution of each of these components.

(1) **Without homophily** (*w/o homophily*). We exclude the homophily loss $\mathcal{L}_i^{\text{homo}}$ from our learning objective. We can see that the team performance drops significantly, especially in tasks with more agents. Moreover, without homophily loss, our method performs worse than all baselines on `Cleanup` (n=5) and (n=10). These observations suggest that our method works mainly because of the homophily loss. (2) **Continuous incentivizing actions**. Ablation *Cont. inc. actions* shows the influence of discrete actions. We learn a continuous incentivizing rewarding function for *w/o homophily*. We can observe further performance decrease on `Cleanup` (n=3) and `Harvest` (n=10). We hypothesize that this is because the search space for incentivizing policies grows, making the strategy more difficult to learn. (3) **Train incentivizing Q's with received incentives**. We ignore incentives received by agents when training incentivizing strategies. For comparison, ablation *w/ inc.* shows what would happen if they are included. We can observe that *w/ inc.* significantly underperforms the original method on all environments. The reason is that agents learn to give each other positive incentives excessively regardless of observations. Since received incentives are also considered in Q-learning for acting behaviors, excessive incentives would overwhelm environmental rewards and significantly hurt learning performance.

## 5.4 EVOLUTION OF COOPERATION

To clearly show the evolution of emergent cooperation, the problem of 2nd-SDs, and how homophily alleviates this problem, according to different team returns, we select four stages during learning and analyze their corresponding behaviors. The detailed stage partition can be found in Fig. 6.

*Phase 1: Exploring incentives*. During this stage, agents are learning basic dynamics of the environment. For example, as shown in the first row of Fig. 6, agents have not learned to eat apples to get rewarded. Meanwhile, during exploration, agents occasionally give incentives to agents who cleaned wastes ($t_1 + 3$), which enables learning incentivizing behaviors in the following stages.

*Phase 2: Second-order social dilemmas*. Some agents (the *pink* agent in the second row of Fig. 6) learn to give positive rewards to cleaning agents altruistically. However, other agents (e.g., *orange* and *blue*) typically do not give positive incentives but can enjoy the benefits of others' altruistic behaviors. We can observe an oscillation of team return during this stage. This is the effect of 2nd-SDs. If there are no additional restrictions (such as homophily) to deal with this problem, the team will fall back to the state where no one wants to perform incentives, resulting in the collapse of cooperation.

*Phase 3: Homophily solves 2nd-SDs*. After some oscillations of cooperation, the homophily loss gradually encourages agents with similar environmental behaviors to have similar incentivizing behaviors. As shown in the third row of Fig. 6, although there are still some noisy incentives at this stage, agents who are close to the apple-spawning region simultaneously reward cleaning agents and punish those who are next to the wastes but do not clean them. These incentivizing behaviors indicate that the population has gotten over the 2nd-SD with the help of homophily. Correspondingly, the team return increases in this stage.

*Phase 4: Stabilized cooperation*. Taking advantage of the effect of homophily, during this stage, there are no second-order free-riders and all incentive rewards are given by agents harvesting apples to agents in the region of wastes. Moreover, screenshots show that agents learn an efficient division of labor – three agents eat apples and get environmental rewards while two agents clean the wastes and get rewards from harvesting agents. These incentivizing and environmental behaviors are a stable solution only when homophily learning objective is included.

Based on the illustrations of the evolution of cooperation, we conclude that homophily prevents altruistic incentivizing behaviors from being exploited by second-order free riders, and thus solves the problem of 2nd-SDs and leads to stable cooperation. For detailed agent behaviors at different stages, we refer readers to our online videos[1].

## 5.5 HOMOPHILIC INCENTIVES

In previous sections, we show that homophilic incentives can promote cooperation, but how homophily affects incentivizing behaviors remains largely unclear. To make up for this, we compare our method with the ablation *w/o homophily* on `Cleanup` ($n$=3) by plotting their collective return and altruistic incentive (the average incentive that each *Cleaner* receives at each time step) in Fig. 7.

We can observe a close connection between incentives and cooperation performance. Intuitively, the more positive incentives cleaners receive, the more apples are expected to spawn and be collected. However, without homophily, the received incentives oscillate dramatically, which is caused by second-order social dilemmas and is in line with the discussions before. In comparison, our method keeps incentivizing cleaners and thus learn cooperation with stability.

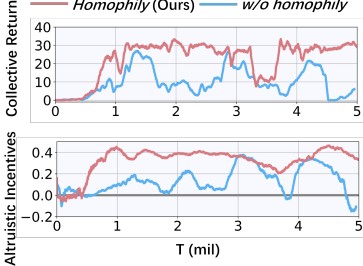

Figure 7: Homophily eliminates incentive oscillations.

## 6 CLOSING REMARKS

In this paper, we study the problem of cooperation emergence. We show that altruistic incentives make cooperation possible but cannot stabilize due to second-order social dilemmas. We then formally and empirically show that homophily, a common tendency typical of humans, may solve this problem. Combined with deep MARL, we propose an implementation of homophilic learning for sequential social dilemmas. We expect that our work can encourage future works on studying the exciting topics of cooperation emergence, evolution, and stability.

_______________

[1]`https://sites.google.com/view/homophily/`

**Reproducibility** The source code for all the experiments along with a README file with instructions on how to run these experiments is attached in supplementary material. In addition, the settings and parameters for all models and algorithms mentioned in the experiment section are detailed in Appendix C.

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

## A    DETAILS OF MOTIVATING EXAMPLE

### A.1    PROBLEM SETTINGS

From the perspective of multi-agent reinforcement learning, we provide a detailed motivating example to explain both first- and second-order social dilemmas, and demonstrate that homophily can promote cooperation by alleviating second-order social dilemmas. The motivating example is based on the classic problem setting from Fowler (2005) for its modeling of the *public goods dilemma*.

A population of $n$ agents has an opportunity to create a public good that will be equally distributed to all participants. There are several possible strategies in this population. Contributors ($C$) pay a cost $c$ to increase the size of public goods by $b$. Defectors ($D$) do not contribute but benefit from the public goods. Agents can also choose to neither contribute to nor benefit from the public goods, but receive a fixed reward $\sigma$, as Nonparticipants ($N$). To introduce altruistic incentives into the system, we add Punishers ($P$) as the fourth type of strategy. A punisher contributes to and benefits from the public goods, which is the same as contributors, and importantly, it also pays a cost $k$ to incur a punishment $p$ on defectors. To guarantee that no agent can exploit altruistic incentives by being a pure contributor that does not spend energy punishing defectors, we consider unexploitable punishers ($PA$) who also pay a cost $\alpha k$ to incur a punishment $\alpha p$ on the pure contributors for not punishing defectors.

In Sec. 3 Part A, to showcase the influence of first-order social dilemmas, we consider Contributors ($C$), Defectors ($D$), and Nonparticipants ($N$). Assume that the proportion of each strategy is $p_C$, $p_D$, and $p_N$, respectively, and $p_C + p_D + p_N = 1$. Then the rewards at each step are

$$r = \begin{cases} \frac{bp_C}{1-p_N} - c, & \text{Contributors } (C) \\ \frac{bp_C}{1-p_N}, & \text{Defectors } (D) \\ \sigma, & \text{Nonparticipants } (N) \end{cases} \tag{7}$$

From Eq. 7 we can see that the rewards of defectors are always higher than those of contributors, which leads to first-order social dilemmas (1st-SDs or SDs).

In Sec. 3 Part B, to show the effect of unexploitable altruistic incentives, we consider Contributors ($C$), Defectors ($D$), Nonparticipants ($N$), and Unexploitable Punishers ($PA$). Assume that the proportion

of each strategy is $p_C$, $p_D$, $p_N$, and $p_{PA}$, respectively, and $p_C + p_D + p_N + p_{PA} = 1$. Then the rewards at each step are

$$
r = \begin{cases}
\frac{b(p_C + p_{PA})}{1 - p_N} - c - \alpha p p_{PA}, & \text{Contributors } (C) \\
\frac{b(p_C + p_{PA})}{1 - p_N} - p p_{PA}, & \text{Defectors } (D) \\
\sigma, & \text{Nonparticipants } (N) \\
\frac{b(p_C + p_{PA})}{1 - p_N} - c - k p_D - \alpha k p_C, & \text{Unexploitable Punishers } (PA)
\end{cases}
\tag{8}
$$

In Sec. 3 Part C, we restrict the punishment incurred by punishers, and they can only pay a cost $k$ to incur a punishment $p$ on defectors. Therefore, in this part, we consider Contributors ($C$), Defectors ($D$), Nonparticipants ($N$), and Exploitable Punishers ($P$). Assume that the proportion of each strategy is $p_C$, $p_D$, $p_N$, and $p_P$, respectively, and $p_C + p_D + p_N + p_P = 1$. Then the rewards each step are

$$
r = \begin{cases}
\frac{b(p_C + p_P)}{1 - p_N} - c, & \text{Contributors } (C) \\
\frac{b(p_C + p_P)}{1 - p_N} - p p_P, & \text{Defectors } (D) \\
\sigma, & \text{Nonparticipants } (N) \\
\frac{b(p_C + p_P)}{1 - p_N} - c - k p_D, & \text{Exploitable Punishers } (P)
\end{cases}
\tag{9}
$$

From Eq. 9, we can find that the rewards of pure contributors are always higher than those of exploitable punishers, which leads to the second-order social dilemmas (2nd-SDs).

In Sec. 3 Part D, to show the effect of homophily, based on the settings of Eq. 9 which suffer from 2nd-SDs, we further encourage agents with similar acting behaviors to have similar incentivizing behaviors. In the considered setting, only contributors and punishers have the same acting behavior of contributing to the public goods. Therefore, we encourage their incentivizing behaviors to be the same by converting the minority of punishers and contributors to the majority of them with probability 0.2. (For example, if 30% of the agents are punishers and 20% of them are contributors, then contributors become punishers with probability 0.2.)

In all our simulations, we set $n = 10$, $b = 3$, $c = 1$, $\sigma = 1$, $p = 2$, $k = 0.35$, and $\alpha = 1$. A sensitivity analysis is provided in Sec. 3 of the main text to demonstrate that our observation and conclusion are solid for a large range of parameters. Agents use REINFORCE policy gradients (Williams, 1992) to learn independent local policies for selecting strategies (atomic actions).

## A.2 Visualization and Coordinate transformation

In order to demonstrate the evolution of population, we employ two visualization methods, the ternary plot (for $p_C, p_D, p_N$ in Sec. 3 Part A) and quaternary plot (for $p_C, p_D, p_N, p_{PA}$ in Sec. 3 Part B and $p_C, p_D, p_N, p_P$ in Sec. 3 Part C, D).

**Ternary plot**, an equilateral triangle periphery with several trajectories inside, reveals the dynamics of population constituents (three different strategies), as shown in Fig. 1(c) in the paper. Each point $X$ inside the equilateral triangle represents a distribution of the population $(p_C, p_D, p_N)$, where $p_C + p_D + p_N = 1$. $p_C$, $p_D$, and $p_N$ is represented by the distance from point $X$ to the edge $ND$, $CN$, and $CD$, respectively. This corresponds to the fact that the sum of distances from any interior points of an equilateral triangle to three sides is a fixed value. Naturally, we can then depict the evolution of the percentage of three strategies as a curve inside the triangle.

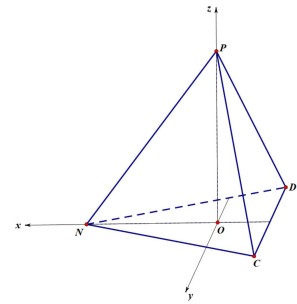

Figure 8: Population policy space in the 3-dimensional rectangular coordinate system.

However, ternary plot can only visualize the evolutionary trajectories of three strategies. When the fourth strategy is introduced, we resort to the **quaternary plot** in 3-dimensional space. For any interior point $X$ of a regular tetrahedron $CDNP$ (Fig. 8), the sum of distances from $X$ to four faces $NCD$, $PCN$, $DPN$, and $PCD$ is a constant. Therefore, we can still use an interior point $X$ to

represent a population distribution and represent $p_C, p_D, p_N, p_P$ by the distances from $X$ to different faces.

Point-to-face distances have to be transformed into rectangular coordinates before trajectories can be plotted. We now describe the details of coordinate transformation.

We first plot a regular tetrahedron in the three-dimensional rectangular coordinate system, with four vertices $P(0,0,1)$, $N(\frac{\sqrt{2}}{2},0,0)$, $C(-\frac{\sqrt{2}}{4}, \frac{\sqrt{6}}{4}, 0)$, $D(-\frac{\sqrt{2}}{4}, -\frac{\sqrt{6}}{4}, 0)$ as in Fig. 8. To simplify notations, we denote the distance from an interior point $X$ to faces $DNP, CNP, CDP$, and $CDN$ by $p_C$, $p_D$, $p_N$, and $p_P$, respectively.

It can be calculated that the equation for plane $PNC$ is

$$\sqrt{2}x + \sqrt{6}y + z = 1$$

and the equation for plane $PND$ is

$$\sqrt{2}x - \sqrt{6}y + z = 1.$$

Next, using formula in analytical geometry for calculating the distance from a point to a given plane, we can derive

$$\begin{cases} p_D = \frac{1}{3}|\sqrt{2}x + \sqrt{6}y + z - 1|, \\ p_C = \frac{1}{3}|\sqrt{2}x - \sqrt{6}y + z - 1|, \\ p_P = z. \end{cases} \tag{10}$$

Finally, we solve the above system of equations and get

$$\begin{cases} x = \frac{\sqrt{2}}{4}(2 - 3p_D - 3p_C - 2p_P), \\ y = \frac{\sqrt{6}}{4}(p_D - p_C), \\ z = p_P, \end{cases} \tag{11}$$

With Eq. 11, we can calculate the corresponding coordinate in the three-dimensional rectangular coordinate system of any population distribution. Connecting all points into a curve gives the 3D quaternary plot.

## B    MORE RESULTS AND ANALYSES

### B.1    MOTIVATING EXAMPLE: LIMIT CYCLE

In Fig. 2 (d) of the paper, we visualize the gradient field within the population policy space when exploitable punishers are considered. There, we conclude that independent REINFORCE learners can never converge to cooperative solutions in this setting. As a special case, we show that a population may rotate infinitely in the non-cooperative region of the policy space.

In Fig. 9, we present a *limit cycle* in the population policy space. A limit cycle is a closed trajectory, and any population initialized on the cycle would keep looping over it and never converge to a stable solution. It is worth noting that although homophilic incentives make cooperation possible (Fig. 3(b) in the main text), they cannot eliminate limit cycles in the non-cooperative (blue) region.

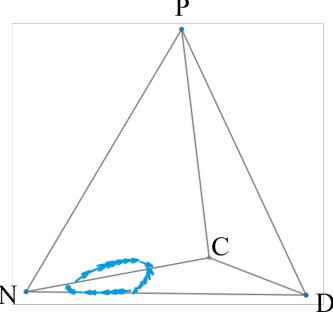

Figure 9: Limit Cycle.

### B.2    MOTIVATING EXAMPLE: DEGREE OF HOMOPHILY

In Sec. 3 Part D, we encourage agents with similar acting behaviors to have similar incentivizing behaviors by converting the minority of punishers and contributors to the majority of them with probability $\lambda$ ($\lambda$=0.2 in the paper). We call $\lambda$ the *degree of homophily*.

To further show the influence of $\lambda$, in Fig. 10, we plot how the *cooperation proportion* changes with respect to the degree of homophily. Cooperation proportion is the ratio of the volume of the

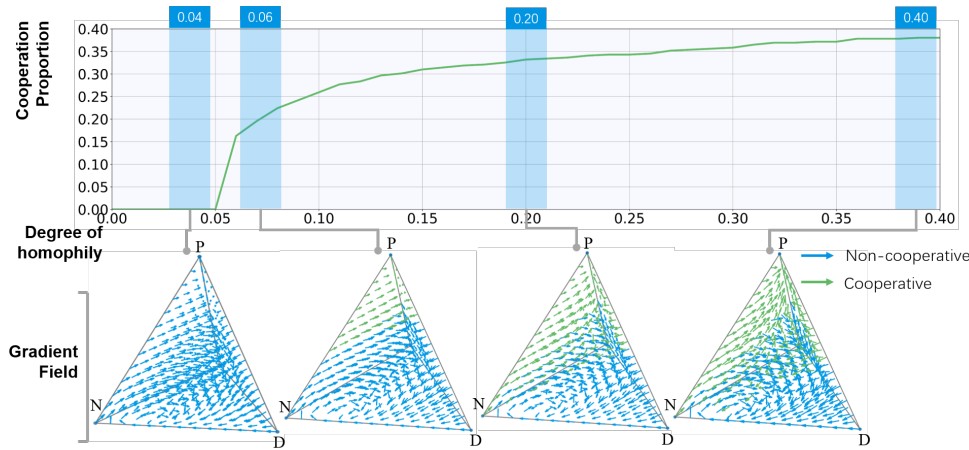

Figure 10: The more homophilic a population is, the more possible cooperation would emerge. The cooperation proportion and gradient field in the population policy space under different degrees of homophily are shown. The increasing volume of the "safe region" and the cooperation proportion increases with the degree of homophily.

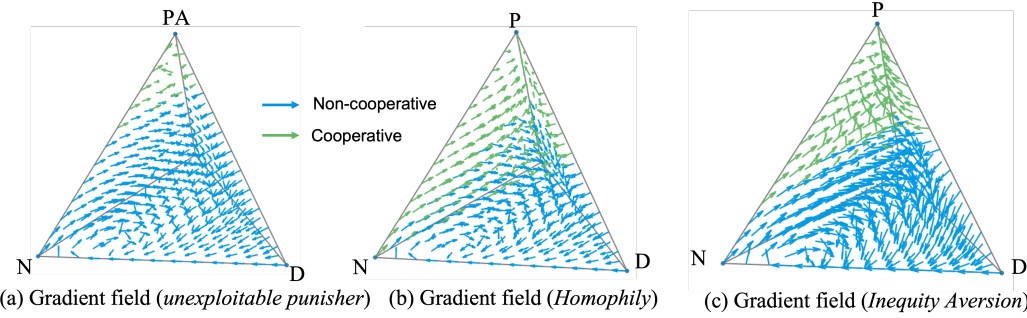

(a) Gradient field (*unexploitable punisher*)    (b) Gradient field (*Homophily*)    (c) Gradient field (*Inequity Aversion*)

Figure 12: Gradient fields of our method against baselines

cooperative (green) region to the volume of the regular tetrahedron. We observe that the cooperation proportion increases with the degree of homophily, and gradually converges to a constant value 0.4. We further present the gradient field under several degrees of homophily. As expected, the volume of the "safe region" increases with the degree of homophily. These results demonstrate that the more homophilic a population is, the more possible cooperation would emerge.

### B.3    MOTIVATING EXAMPLE: SENSITIVITY ANALYSIS

We use one-at-a-time method and show the result with a Tornado Diagram in Fig. 11. When homophily is *not* introduced, with a 20% perturbation of default parameters, 2nd-SDs persist and the "cooperative volume" (green region in Fig. 2(d)) is always 0 (Fig. 11 left). In contrast, when homophily is introduced, the cooperative volume (green region in Fig. 3(b)) fluctuates within a reasonable range and is always larger than 0.1 (Fig. 11 right), which allows cooperation emergence. These results consolidate the robustness of our observations and method across a wide range of conditions.

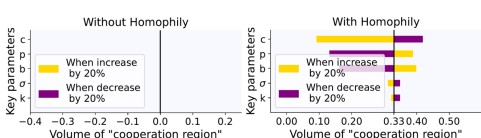

Figure 11: Sensitivity analysis.

### B.4    MOTIVATING EXAMPLE: GRADIENT FIELD OF INEQUITY AVERSION

In Sec. 3, we use the gradient field plot to show the effect of social dilemmas and homophily. Here to demonstrate the mechanism behind other baselines, we plot the gradient field of Inequity

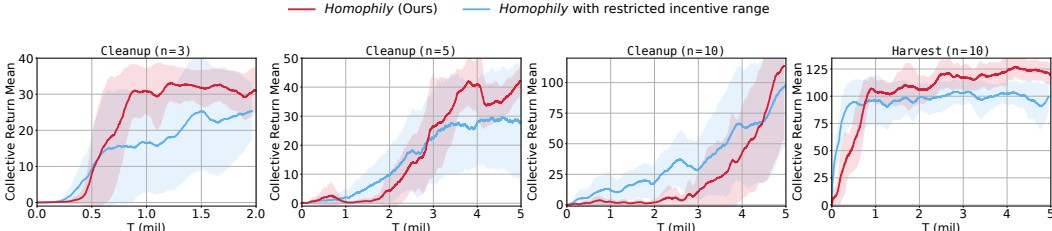

Figure 13: Performance of our method when the incentive range is restricted

Aversion (Hughes et al., 2018). We adopt the inequity aversion model used in its original paper as in Equ. 12.

$$r_i^{\text{inequity aversion}} = -\frac{\alpha}{N-1} \sum_{j \neq i} \max(r_j - r_i, 0) - \frac{\beta}{N-1} \sum_{j \neq i} \max(r_i - r_j, 0) \qquad (12)$$

where $\alpha = 5$ and $\beta = 0.05$ in their default settings.

We compare the gradient field of our method and other baselines in Fig. 12. The cooperative area volume of inequity aversion is smaller than our method (about 60%). Moreover, unlike our method, there is no cooperative area around the vertex N. Once the population of agents is trapped in the plane C-D-N, they can never escape from it.

## B.5  SSDs: Restricted Incentive Range

In Sec. 4, we allow the agents to incentivize any other agents even if they are not in the view range. Here we consider restricting the incentive range to the view size. We compare this restricted variant with the original unrestricted version in Fig. 13. There is only a slightly performance decrease in the restricted version, which shows the robustness of our method.

## B.6  SSDs: Fairness in Cleanup

In Sec. 5.4, we have observed the division of labor between agents in `Cleanup` (n=5). In real life, fairness is a very important indicator of how good the division of labor is. Here we statistically measure the fairness of the system, and find that the agents carrying out different tasks are roughly fair in `Cleanup` (n=5), when we take the incentivizing rewards into consideration. We use the Equality metric (Hughes et al., 2018) to measure the fairness:

$$E = 1 - \frac{\sum_{i=1}^{n} \sum_{j=1}^{n} |r_i - r_j|}{2n \sum_{i=1}^{n} r_i} \qquad (13)$$

where $r_i$ is agent $i$'s reward and $n$ is the number of agents.

The value of equlity metric is between 0 and 1. The closer the value to 1, the fairer the system is. We measure the fairness of models learned with 5 random seeds and calculate the mean value as well as 95% confidence intervals. The mean values and confidence intervals of equality when we only consider environmental rewards and when we consider both environmental and incentivizing rewards are 0.6693(0.1252) and 0.8900(0.0473), respectively. We conjecture that the fairness in this system comes from the fact that apple-harvesters and waste-cleaners should have roughly equal total rewards, otherwise no agents would choose to clean wastes.

## B.7  Results of Different Incentive Coefficients

For single-stage social dilemmas, we have conducted a sensitivity analysis in Sec. 3, which shows our method are quite robust to these hyperparameters. For SSDs, we also observe similar results. However, when extreme values of these hyperparameters are used, the performance can severely

degrade. Here we show the results when we change the incentive effect factor $\eta^e$, and incentive cost factor $\eta^c$ in the following table.

Table 1: Results of Different Incentive Coefficients.

| Configs ($\eta^e$,$\eta^c$) | Team Return | Incentivizing Action Frequency | Incentives for Cleaning Waste |
|---|---|---|---|
| (1.0, 0.1)(Ours) | **46.80(2.85)** | 0.1165(0.0002) | 0.800(0.020) |
| (1.0, 1.0) | 3.46(3.29) | 0.0334(0.0004) | -0.001(0.013) |
| (1.0, 0.0) | 8.56(6.72) | 0.4741(0.0009) | 0.440(0.200) |
| (0.0, 0.0) | 2.80(1.34) | - | - |

Each item in the above table is formatted with *mean(std)*. When the incentive cost $\eta^c$ is equal to incentive effect $\eta^e$, agents tend to not use incentivizing actions, which makes it difficult to achieve cooperation. On the contrary, when the incentive cost $\eta^c$ is zero, agents use incentivizing actions too frequently, which interferes with the learning of environmental policies.

### B.8 INTUITION OF HOMOPHILY

We have provided intuition for introducing homophily by our motivating example in Sec. 3. Here we provide more explanations of why homophily is an effective heuristic by (1) using a real-world example and (2) giving more explanations about the impact of homophily in SSDs in the experiment section.

(1) In a real-world situation where there are multiple social infrastructure builders and taxpayers, the taxpayers' benefits depend on the builders' contribution and the builders' profits depend on gathered taxes. In this case, taxpayers should have similar tax payment strategies. Otherwise, more and more taxpayers will choose to be free-riders and refuse to pay the tax, then the cooperation will collapse.

(2) In the evolution of cooperation of Cleanup in Sec. 5.4, apple-harvesters under the constraint of homophily tend to associate with other apple-harvesters, and tend not to be a second-order free-rider, *i.e.*, refusing to give positive rewards to waste-cleaners. Then the altruistic apple-harvesters will not be exploited by second-order free-riders and the waste-cleaners will keep receiving rewards while cleaning. Thus the emergent cooperation becomes stable.

### B.9 DISCUSSION

**Is homophily the only possible solution for 2nd-SDs?** No, it is not, but homophily has several advantages over other possible solutions. There are three types of solutions for (1st-) SDs (Kollock, 1998). (1) *Motivational* solutions assume that agents take their partners' rewards into account when making a decision (Kuhlman & Marshello, 1975), which build in a strong bias towards cooperation. (2) *Strategic* solutions, such as tit-for-tat (Axelrod & Hamilton, 1984) and partner selection (Anastassacos et al., 2020), typically require identifying cooperative and defective behaviors. Such approaches face major challenges in temporally extended cases – the co-learning of cooperation/defection detection and incentivizing mechanism is fragile and suffers from extremely postponed learning signals from each other. (3) *Structural* solutions modify the game settings (Rapoport et al., 1989) and thus may violate the assumption of the study of cooperation emergence. The advantage of homophily is that it avoids higher-order SDs (e.g., 3rd-SDs) while not requiring identifying cooperation/defection.

## C DETAILS OF EXPERIMENTS

Our method is built on the open-sourced codebase PyMARL (Samvelyan et al., 2019) and Sequential Social Dilemma Games (SSDG) (Vinitsky et al., 2019). We now discuss the specific environmental settings and implementation details.

### C.1 SEQUENTIAL SOCIAL DILEMMAS

In `Cleanup` game (Fig. 14(a)), the apple regrowth rate decreases linearly with the density of waste and would reduce to zero as the amount of waste exceeds a saturation threshold. At the start of each episode, the environment resets with waste just beyond the saturation threshold. To enable apple reproduction, agents can fire a cleaning beam to clear waste within the beam range, but this action induces no immediate reward. Additionally, waste is produced with a certain probability during the episode, so agents need to keep cleaning waste to maintain the provision of apples. We have a dilemma in this game. There is a distance between the waste field and the apple field. It is personally more rewarding to stay in the apple field to wait for the spawned apples, but some agents have to sacrifice, leave the apple field, and contribute to the public goods by cleaning waste.

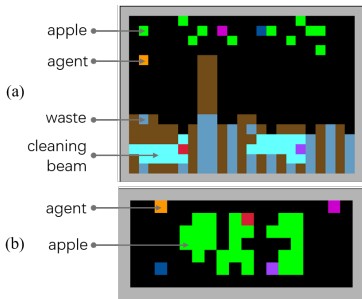

Figure 14: (a) `Cleanup` and (b) `Harvest` Game.

In `Harvest` game (Fig. 14(b)), the apple reproduction rate grows with the density of uncollected apples around it. If all apples in a local area are harvested then no apple will re-spawn until the next episode. There is a dilemma in the game. For individual short-term interests, agents tend to harvest as rapidly as possible, which will deplete local resources and harm long-term collective interests. However, agents who abstain from personal benefits for the good of the group are easily exploited by defecting and over-harvesting agents.

## C.2 ENVIRONMENTS

To focus on the problem of social dilemmas, for all experiments, including baselines and ablations, we remove agent rotation actions in SSDG and set the orientation of all agents to face "up", like in previous works studying social dilemmas (Yang et al., 2020). For our method, we disable the "fining" action and use incentive actions as the way to incentivize other agents. In both `Cleanup` and `Harvest` game, eating an apple will provide a local reward of +1, and there are no other environmental rewards.

In `Cleanup`, agents are equipped with a cleaning beam, which allows them to remove waste. We test our method on three `Cleanup` maps with different numbers of agents. In Table 2, we show the details of each map.

Table 2: Environmental settings for `Cleanup` with different numbers of agents.

| Parameter | n=3 | n=5 | n=10 |
|---|---|---|---|
| Map Size | 10×10 | 25×18 | 48×18 |
| View Size | 7 | 7 | 7 |
| Max Steps | 50 | 100 | 100 |
| Apple Respawn Probability | 0.3 | 0.05 | 0.05 |
| Depletion Threshold | 0.4 | 0.99 | 0.99 |
| Restoration Threshold | 0.0 | 0.0 | 0.0 |
| Waste Spawn Probability | 0.5 | 0.05 | 0.05 |

In `Harvest`, the apple spawning rate at each point is related to the current number of apples within an $\ell_1$-distance of 2, and the spawn probability is 0, 0.05, 0.08, and 0.1 when there are 0, 1, 2, and $\geq 3$ apples within the distance, respectively.

## C.3 IMPLEMENTATION

**Network architecture** There are two main components in our methods: environmental Q-functions and incentivizing Q-functions. In the study of social dilemmas, agents learn independently, so we do not share parameters among different agents, except for the RGB preprocessing network, which has the following architecture: 1 convolutional layer (6 filters, 3× 3 kernel with stride 1), 1 flatten layer, and 1 dense layer (32 neurons) with LeakyReLU activation. The environmental Q-functions and incentivizing Q-functions take the output of the RGB preprocessing network as input.

The environmental Q-function consists of three layers: a fully-connected layer with LeakyReLU as activation, followed by a 64 bit GRU, and followed by another fully-connected layer that outputs an estimated utility for each action. The incentivizing Q-function has the same architecture as the acting Q-function, except that the last layer additionally takes the target agent's environmental action as input.

**Optimization** For all experiments, we set the loss scaling factor $\lambda^{\text{inc}} = 1$, $\lambda^{\text{homo}} = 0.01$, discount factor $\gamma^{\text{env}} = 0.95$, $\gamma^{\text{inc}} = 0.995$, incentive effect factor $\eta^{\text{e}} = 1.0$, and incentive cost factor $\eta^{\text{c}} = 0.1$. The optimization is conducted using Adam with a learning rate of 0.0001. For exploration, we use $\epsilon$-greedy with $\epsilon$ annealed linearly from 1.0 to 0.05 over 50K time steps and kept constant for the rest of the training. We run one environment each time to collect samples. Batches of 16 episodes are sampled from the replay buffer (whose size is 5000), and the whole framework is trained end-to-end on fully unrolled episodes.

Experiments are carried out on NVIDIA GTX 2080 Ti GPU. For a 10-agent environment (like `Cleanup` ($n$=10)), our method requires approximately 60G of RAM and 0.9G of video memory, and takes about 115 hours to finish 5M timesteps of training.

### C.4 Similarity of environmental behaviors

In Sec. 4.2 of the paper, we discuss how to encourage homophily in temporally extended cases. A challenge is to measure the similarity of environmental behaviors. In this paper, we use the X-means (Pelleg et al., 2000) implementation provided by PyClustering (Novikov, 2019) to identify similar behaviors. The behavioral features include the changes made to common information like the amount of harvesting, the amount of cleanup, etc., in the last 10 timesteps. When clustering, the minimum and maximum number of clusters are set to 2 and 4, respectively.

## D Details of baselines and ablations

We compare our method against various baselines and ablations. For LIO (Yang et al., 2020), Inequity Aversion (Hughes et al., 2018), and Social Influence (Jaques et al., 2019), we use the codes provided by the authors and the hyper-parameters that have been fine-tuned on the Sequential Social Dilemma Games (SSDG) (Vinitsky et al., 2019). For Selfish Actor-Critic, we use the default implementation in SSDG.

The ablation *Cont. inc. actions*, which learns continuous incentives, uses the same implementation and hyperparameter settings as LIO, but with the only difference that the incentive cost is changed from the default value of LIO to 0.1, which is the same as in our method. Ablation *w/o homophily* and *w/ inc* use the identical network structure and hyper-parameter settings as our method. The differences are that *w/o homophily* does not use the homophily loss by setting homophily loss scaling factor $\lambda^{\text{homo}}$ to 0, and *w/ inc* additionally uses received incentives to train incentive Q-functions. For all the experiments, we keep the game settings to be the same for fair comparisons.

## E Formal Analysis

In this section, we formally calculate the closed-form gradients in the policy space for the motivating example problem, which are visualized as ternary plots or quaternary plots in Sec. 3 of the paper.

### E.1 Notations

Suppose a team of $n$ agents joins the *public goods game* introduced in Sec. 3, where the reward functions can be found in Appendix A. The action set $A$ is identical for all agents and depends on the specific settings in the following subsections. Agent $i$ chooses actions according to its own policy $\pi_i$, which is parameterized by $|A|$ parameters, and $\sum_{X \in A} \pi_i(a_i = X) = 1$. In addition, we assume agents choose actions independently, that is $\pi(a_1, \cdots, a_n) = \prod_{j=1}^{n} \pi_j(a_j)$.

We first define two frequently used equations. Let $\pi_k(a_k = N) = \theta_{k,N}$, where $N$ represents the action of non-participating, $\theta_{k,N} \in [0, 1]$, and $k = 1, 2, \cdots n$.

First we define

$$E_{-i} = \mathbb{E}_{\pi} \left[ \frac{1}{n - \sum_{k \neq i} \mathbb{I}(a_k = N)} \right],$$

where $\sum_{k \neq i} \mathbb{I}(a_k = N)$ is the number of agents taking the non-participating action $N$. We use the notation $\mathbf{d}_{-i} = (d_1, \cdots, d_{i-1}, d_{i+1}, \cdots, d_n)$, where $d_k \equiv \mathbb{I}(a_k = N) \in \{0, 1\}$ is the indicator variable, i.e., $d_k = 1$ if $a_k = N$ and otherwise $d_k = 0$. We have

$$E_{-i} = \sum_{\mathbf{d}_{-i} \in \{0,1\}^{n-1}} \frac{1}{n - \sum_{k \neq i} d_k} \prod_{k \neq i} (1 - \theta_{k,N})^{1-d_k} \theta_{k,N}^{d_k}. \tag{14}$$

Note that $\frac{\partial}{\partial \theta_{i,N}} E_{-i} = 0$. Similarly, we define

$$E_{-ij} = \mathbb{E}_{\pi} \left[ \frac{1}{n - \sum_{k \notin \{i,j\}} \mathbb{I}(a_k = N)} \right]$$

and use the notation $\mathbf{d}_{-ij} = (d_1, \cdots, d_{i-1}, d_{i+1}, \cdots, d_{j-1}, d_{j+1}, \cdots, d_n)$, where $d_k \equiv \mathbb{I}(a_k = N) \in \{0, 1\}$. We have

$$E_{-ij} = \sum_{\mathbf{d}_{-ij} \in \{0,1\}^{n-2}} \frac{1}{n - \sum_{k \notin \{i,j\}} d_k} \prod_{k \notin \{i,j\}} (1 - \theta_{k,N})^{1-d_k} \theta_{k,N}^{d_k}. \tag{15}$$

Note that $\frac{\partial}{\partial \theta_{i,N}} E_{-ij} = 0$.

## E.2 Sec. 3 Part A

To showcace the influence of first-order social dilemmas, in Sec. 3 Part A we consider three atomic actions, $C, D, N$. Assume that agent $i$ chooses actions from $A = \{C, D, N\}$ with probabilities $\theta_{i,C}, \theta_{i,D}, \theta_{i,N}$. The reward function is defined in Eq. 7 in Appendix A.1. We can calculate the value function of agent $i$:

$$
\begin{aligned}
V_i^{\pi} =& \mathbb{E}_{\pi} \left[ r_i(a_1, \cdots, a_n) \right] \\
=& \mathbb{E}_{\pi} \left[ \mathbb{I}(a_i = C) \left( b \frac{1}{n - \sum_{j \neq i} \mathbb{I}(a_j = N)} \left( \sum_{j \neq i} \mathbb{I}(a_j = C) + 1 \right) - c \right) \right. \\
& + \mathbb{I}(a_i = D) b \frac{1}{n - \sum_{j \neq i} \mathbb{I}(a_j = N)} \left( \sum_{j \neq i} \mathbb{I}(a_j = C) \right) \\
& \left. + \mathbb{I}(a_i = N) \sigma \right] \\
=& \theta_{i,C} \left( b \left( \sum_{j \neq i} \theta_{j,C} E_{-ij} + E_{-i} \right) - c \right) + \theta_{i,D} b \sum_{j \neq i} \theta_{j,C} E_{-ij} + \theta_{i,N} \sigma,
\end{aligned}
$$

where $E_{-i}$ and $E_{-ij}$ are defined as in Eq. 14 and Eq. 15, respectively. Then we can calculate the derivatives of the value function w.r.t each parameter:

$$\frac{\partial V_i^{\pi}}{\partial \theta_{i,C}} = b \left( \sum_{j \neq i} \theta_{j,C} E_{-ij} + E_{-i} \right) - c, \tag{16}$$

$$\frac{\partial V_i^\pi}{\partial \theta_{i,D}} = b \sum_{j \neq i} \theta_{j,C} E_{-ij}, \tag{17}$$

$$\frac{\partial V_i^\pi}{\partial \theta_{i,N}} = \sigma. \tag{18}$$

The above equations are used to plot Fig. 1(d) of the paper.

### E.3 SEC. 3 PART B

To introduce altruistic incentives into the system, we add the fourth type of atomic action, $PA$. We consider unexploitable punishments here, and exploitable punishments in the next subsection. Agent $i$ chooses actions from $A = \{C, D, N, PA\}$ with probabilities $\theta_{i,C}, \theta_{i,D}, \theta_{i,N}, \theta_{i,PA}$. The reward function in this case is defined in Eq. 8 in Appendix A.1. We can calculate the value function of agent $i$,

$$
\begin{aligned}
V_i^\pi =& \mathbb{E}_\pi \left[ r_i(a_1, \cdots, a_n) \right] \\
=& \mathbb{E}_\pi \left[ \mathbb{I}(a_i = C) \left( b \frac{1}{n - \sum_{j \neq i} \mathbb{I}(a_j = N)} \left( \sum_{j \neq i} \mathbb{I}(a_j = C, PA) + 1 \right) - c \right. \right. \\
& \left. - \alpha p \frac{1}{n} \sum_{j \neq i} \mathbb{I}(a_j = PA) \right) \\
& + \mathbb{I}(a_i = D) \left( b \frac{1}{n - \sum_{j \neq i} \mathbb{I}(a_j = N)} \left( \sum_{j \neq i} \mathbb{I}(a_j = C, PA) \right) - p \frac{1}{n} \sum_{j \neq i} \mathbb{I}(a_j = PA) \right) \\
& + \mathbb{I}(a_i = N)\sigma \\
& + \mathbb{I}(a_i = PA) \left( b \frac{1}{n - \sum_{j \neq i} \mathbb{I}(a_j = N)} \left( \sum_{j \neq i} \mathbb{I}(a_j = C, PA) + 1 \right) - c \right. \\
& \left. \left. - \alpha k \frac{1}{n} \sum_{j \neq i} \mathbb{I}(a_j = C) - k \frac{1}{n} \sum_{j \neq i} \mathbb{I}(a_j = D) \right) \right] \\
=& \theta_{i,C} \left( b \left( \sum_{j \neq i} (\theta_{j,C} + \theta_{j,PA}) E_{-ij} + E_{-i} \right) - c - \alpha \frac{1}{n} p \sum_{j \neq i} \theta_{j,PA} \right) \\
& + \theta_{i,D} \left( b \sum_{j \neq i} (\theta_{j,C} + \theta_{j,PA}) E_{-ij} - p \frac{1}{n} \sum_{j \neq i} \theta_{j,PA} \right) \\
& + \theta_{i,N}\sigma \\
& + \theta_{j,PA} \left( b \left( \sum_{j \neq i} (\theta_{j,C} + \theta_{j,PA}) E_{-ij} + E_{-i} \right) - c - \alpha k \frac{1}{n} \sum_{j \neq i} \theta_{j,C} - k \frac{1}{n} \sum_{j \neq i} \theta_{j,D} \right),
\end{aligned}
$$

where $\mathbb{I}(a_j = C, PA)$ equals one if and only if $a_j = C$ or $a_j = PA$, and $E_{-i}$ and $E_{-ij}$ are defined in Eq. 14 and Eq. 15, respectively. Then we can calculate the derivatives of the value function w.r.t each parameter,

$$\frac{\partial V_i^\pi}{\partial \theta_{i,C}} = b \left( \sum_{j \neq i} (\theta_{j,C} + \theta_{j,PA}) E_{-ij} + E_{-i} \right) - c - \alpha p \frac{1}{n} \sum_{j \neq i} \theta_{j,PA}, \tag{19}$$

$$\frac{\partial V_i^\pi}{\partial \theta_{i,D}} = b \sum_{j \neq i} (\theta_{j,C} + \theta_{j,PA}) E_{-ij} - p \frac{1}{n} \sum_{j \neq i} \theta_{j,PA}, \tag{20}$$

$$\frac{\partial V_i^\pi}{\partial \theta_{i,N}} = \sigma, \tag{21}$$

$$\frac{\partial V_i^\pi}{\partial \theta_{i,PA}} = b \left( \sum_{j \neq i} (\theta_{j,C} + \theta_{j,PA}) E_{-ij} + E_{-i} \right) - c - \alpha k \frac{1}{n} \sum_{j \neq i} \theta_{j,C} - k \frac{1}{n} \sum_{j \neq i} \theta_{j,D}. \tag{22}$$

The above equations are used to plot Fig. 3(a) of the paper.

### E.4 Sec. 3 Part C

Now we consider exploitable punishments. Agent $i$ chooses actions from $A = \{C, D, N, P\}$ with probabilities $\theta_{i,C}, \theta_{i,D}, \theta_{i,N}, \theta_{i,P}$. The reward function is defined in Eq. 9 in Appendix A.1. We can calculate the value function of agent $i$:

$$
\begin{aligned}
V_i^\pi =& \mathbb{E}_\pi \left[ r_i(a_1, \cdots, a_n) \right] \\
=& \mathbb{E}_\pi \left[ \mathbb{I}(a_i = C) \left( b \frac{1}{n - \sum_{j \neq i} \mathbb{I}(a_j = N)} \left( \sum_{j \neq i} \mathbb{I}(a_j = C, P) + 1 \right) - c \right) \right. \\
& + \mathbb{I}(a_i = D) \left( b \frac{1}{n - \sum_{j \neq i} \mathbb{I}(a_j = N)} \left( \sum_{j \neq i} \mathbb{I}(a_j = C, P) \right) - p \frac{1}{n} \sum_{j \neq i} \mathbb{I}(a_j = P) \right) \\
& + \mathbb{I}(a_i = N)\sigma \\
& + \mathbb{I}(a_i = P) \left( b \frac{1}{n - \sum_{j \neq i} \mathbb{I}(a_j = N)} \left( \sum_{j \neq i} \mathbb{I}(a_j = C, P) + 1 \right) - c \right. \\
& \left. \left. - k \frac{1}{n} \sum_{j \neq i} \mathbb{I}(a_j = D) \right) \right] \\
=& \theta_{i,C} \left( b \left( \sum_{j \neq i} (\theta_{j,C} + \theta_{j,P}) E_{-ij} + E_{-i} \right) - c \right) \\
& + \theta_{i,D} \left( b \sum_{j \neq i} (\theta_{j,C} + \theta_{j,P}) E_{-ij} - p \frac{1}{n} \sum_{j \neq i} \theta_{j,P} \right) \\
& + \theta_{i,N}\sigma \\
& + \theta_{j,P} \left( b \left( \sum_{j \neq i} (\theta_{j,C} + \theta_{j,P}) E_{-ij} + E_{-i} \right) - c - k \frac{1}{n} \sum_{j \neq i} \theta_{j,D} \right),
\end{aligned}
$$

where $\mathbb{I}(a_j = C, P)$ equals one if and only if $a_j = C$ or $a_j = P$, and $E_{-i}$ and $E_{-ij}$ are defined in Eq. 14 and Eq. 15, respectively. Then we can calculate the derivatives of the value function w.r.t each parameter,

$$\frac{\partial V_i^\pi}{\partial \theta_{i,C}} = b \left( \sum_{j \neq i} (\theta_{j,C} + \theta_{j,P}) E_{-ij} + E_{-i} \right) - c, \tag{23}$$

$$\frac{\partial V_i^\pi}{\partial \theta_{i,D}} = b \sum_{j \neq i} (\theta_{j,C} + \theta_{j,P}) E_{-ij} - p \frac{1}{n} \sum_{j \neq i} \theta_{j,P}, \tag{24}$$

$$\frac{\partial V_i^\pi}{\partial \theta_{i,N}} = \sigma, \tag{25}$$

$$\frac{\partial V_i^\pi}{\partial \theta_{i,P}} = b \left( \sum_{j \neq i} (\theta_{j,C} + \theta_{j,P}) E_{-ij} + E_{-i} \right) - c - k\frac{1}{n} \sum_{j \neq i} \theta_{j,D}. \tag{26}$$

The above equations are used to plot Fig. 2(d) of the paper. It is worth noting that

$$\frac{\partial V_i^\pi}{\partial \theta_{i,P}} - \frac{\partial V_i^\pi}{\partial \theta_{i,C}} = -k\frac{1}{n} \sum_{j \neq i} \theta_{j,D} \leq 0. \tag{27}$$

The gradient is non-positive for any $\theta_{j,D}$, which indicates that second-order cooperators would be taken over by second-order defectors.

### E.5 SEC. 3 PART D

Based on the settings specified in Appendix E.4, we consider homophily here. Agent $i$ chooses actions from $A = \{C, D, N, P\}$ with probabilities $\theta_{i,C}, \theta_{i,D}, \theta_{i,N}, \theta_{i,P}$. The reward function is defined in Eq. 9. We can calculate the value function of agent $i$:

$$
\begin{aligned}
V_i^\pi =& \mathbb{E}_\pi \left[ r_i(a_1, \cdots, a_n) \right] \\
=& \mathbb{E}_\pi \Bigg[ \mathbb{I}(a_i = C) \left( b \frac{1}{n - \sum_{j \neq i} \mathbb{I}(a_j = N)} \left( \sum_{j \neq i} \mathbb{I}(a_j = C, P) + 1 \right) - c \right) \\
& + \mathbb{I}(a_i = D) \left( b \frac{1}{n - \sum_{j \neq i} \mathbb{I}(a_j = N)} \left( \sum_{j \neq i} \mathbb{I}(a_j = C, P) \right) - p\frac{1}{n} \sum_{j \neq i} \mathbb{I}(a_j = P) \right) \\
& + \mathbb{I}(a_i = N)\sigma \\
& + \mathbb{I}(a_i = P) \Bigg( b \frac{1}{n - \sum_{j \neq i} \mathbb{I}(a_j = N)} \left( \sum_{j \neq i} \mathbb{I}(a_j = C, P) + 1 \right) - c \\
& - k\frac{1}{n} \sum_{j \neq i} \mathbb{I}(a_j = D) \Bigg) \Bigg] \\
=& \theta_{i,C} \left( b \left( \sum_{j \neq i} (\theta_{j,C} + \theta_{j,P}) E_{-ij} + E_{-i} \right) - c \right) \\
& + \theta_{i,D} \left( b \sum_{j \neq i} (\theta_{j,C} + \theta_{j,P}) E_{-ij} - p\frac{1}{n} \sum_{j \neq i} \theta_{j,P} \right) \\
& + \theta_{i,N}\sigma \\
& + \theta_{j,P} \left( b \left( \sum_{j \neq i} (\theta_{j,C} + \theta_{j,P}) E_{-ij} + E_{-i} \right) - c - k\frac{1}{n} \sum_{j \neq i} \theta_{j,D} \right),
\end{aligned}
$$

where $\mathbb{I}(a_j = C, P)$ equals one if and only if $a_j = C$ or $a_j = P$, and $E_{-i}$ and $E_{-ij}$ are defined in Eq. 14 and Eq. 15, respectively. Then we can calculate the derivatives of the value function w.r.t each parameter. Besides, to show the effect of homophily, we encourage agents with similar environmental behaviors to have similar incentivizing behaviors. Since only contributors and punishers have the same environmental behavior of contributing to the public goods, we encourage their incentivizing behaviors to be the same by converting the minority of P and C to the majority with a probability of

$\lambda = 0.2$. For each agent, the converting direction depends on all the other agents. Specifically, if other agents choose action $P$ more often ($\text{sign}(\sum_{j \neq i}(\theta_{j,P} - \theta_{j,C})) > 0$), agent $i$ will increase the probability of choosing action $P$ by $\lambda \cdot \min(\theta_{i,C}, \theta_{i,P})$, and vice versa. Formally, taken homophily into consideration, the gradients of each parameter of agent $i$ are:

$$\dot{\theta}_{i,C} = b \left( \sum_{j \neq i}(\theta_{j,C} + \theta_{j,P})E_{-ij} + E_{-i} \right) - c \tag{28}$$

$$+ \lambda \cdot \text{sign} \left( \sum_{j \neq i}(\theta_{j,C} - \theta_{j,P}) \right) \min(\theta_{i,C}, \theta_{i,P}), \tag{29}$$

$$\dot{\theta}_{i,D} = b \sum_{j \neq i}(\theta_{j,C} + \theta_{j,P})E_{-ij} - p \frac{1}{n} \sum_{j \neq i} \theta_{j,P}, \tag{30}$$

$$\dot{\theta}_{i,N} = \sigma, \tag{31}$$

$$\dot{\theta}_{i,P} = b \left( \sum_{j \neq i}(\theta_{j,C} + \theta_{j,P})E_{-ij} + E_{-i} \right) - c - k \frac{1}{n} \sum_{j \neq i} \theta_{j,D} \tag{32}$$

$$+ \lambda \cdot \text{sign} \left( \sum_{j \neq i}(\theta_{j,P} - \theta_{j,C}) \right) \min(\theta_{i,C}, \theta_{i,P}). \tag{33}$$

The above equations are used to plot Fig. 3(b) of the paper. And it is worth noting that

$$\dot{\theta}_{i,P} - \dot{\theta}_{i,C} = -k \frac{1}{n} \sum_{j \neq i} \theta_{j,D} + 2\lambda \cdot \text{sign} \left( \sum_{j \neq i}(\theta_{j,P} - \theta_{j,C}) \right) \min(\theta_{i,C}, \theta_{i,P}), \tag{34}$$

which is positive when the population is close to point P, in which situation the population chooses action $P$ more often than $C$ and $D$ ($\text{sign}(\sum_{j \neq i}(\theta_{j,P} - \theta_{j,C})) > 0$ and $\sum_{j \neq i} \theta_{j,D}$ is small), and thus the agents will not deviate to second-order free-riding.

### E.6 PROJECT ONTO THE PROBABILITY SIMPLEX

After we have obtained the gradients of each action of agent $i$, we can calculate the new policy as follows:

$$\theta'_{i,X} = \theta_{i,X} + \beta \frac{\partial V_i^\pi}{\partial \theta_{i,X}}, \forall X \in A. \tag{35}$$

where $\beta$ is the learning rate. Now $\pi'_i(a_i = X) = \theta'_{i,X}$. However, this new policy may not be in the valid probability space, i.e. $\sum_{X \in A} \pi'_i(a_i = X) = 1$ may not hold anymore. To address this issue, we project it onto the probability simplex (Chen & Ye, 2011; Wang & Carreira-Perpiñán, 2013) after each update. Let $\Pi_{\Delta^{|A|}} : \mathbb{R}^{|A|} \to \Delta^{|A|}$ denotes the projection to the valid space:

$$\Pi_{\Delta^{|A|}}[\boldsymbol{x}] = \arg \min_{\boldsymbol{z} \in \Delta^{|A|}} \|\boldsymbol{x} - \boldsymbol{z}\|, \tag{36}$$

where $\| \cdot \|$ denotes the regular Euclidean norm and $\Delta^{|A|}$ is the canonical simplex defined by

$$\Delta^{|A|} := \left\{ \boldsymbol{z} = (z_1, \cdots, z_{|A|})^\top \in \mathbb{R}^{|A|} : 0 \leq z_i \leq 1, i = 1, \cdots, n, \text{ and } \sum_{i=1}^{|A|} z_i = 1 \right\}.$$

Using this projection, the valid new policy of agent $i$ is

$$\boldsymbol{\pi}_i' \leftarrow \Pi_{\Delta^{|A|}}[\boldsymbol{\pi}_i'], \tag{37}$$

where $\boldsymbol{\pi}_i' = (\theta_{i,X}')_{X \in A}^\top$ is the parameter vector of agent $i$. With enough updates and this projection rule, we can determine the location of the "safe region" in Fig. 2(d), Fig. 3(a) and Fig. 3(b) of the paper.

## F  LIMITATIONS

### F.1  LIMITATIONS OF FORMAL ANALYSIS

In this work, we study the problem of cooperation emergence in both single-stage and sequential social dilemmas. For single-stage social dilemmas, we provide formal analyses and numerical results for the effect of free-riders and homophily in Sec. 3 and Appendix E. However, in the main text, for sequential social dilemmas, we only provide empirical results in Sec. 5.

Our aim is to formally analyze the evolutionary dynamics of environmental behaviors and incentivizing behaviors on SSDs. However, the environmental policies and incentivizing policies are interdependent, and it is difficult to accurately account for the effect of incentivizing behaviors when considering environmental policy updates.

To better explain this challenge, we first give some notations. There are $n$ agents participating in the *public goods game* introduced in Sec. 3. The first-order cooperation contributes to the public goods and benefits from it, while the first-order defection does not contribute but benefits from the public goods. The second-order cooperation will pay a cost to punish those who choose first-order defection, while the second-order defection will not punish such defections. Each agent $i$ chooses actions according to its own policy $\pi_i$, which includes environmental policy $\pi_i^1(a_i^1)$ parameterized by $\theta_{i,1}$ and incentivizing policy $\pi_i^2(a_i^2)$ parameterized by $\theta_{i,2}$. We assume that the two policies are independent, i.e., $\pi_i(a_i^1, a_i^2) = \pi_i^1(a_i^2)\pi_i^2(a_i^2)$. Agent $i$ chooses first-order cooperation (or defection) with probability $\pi_i^1(a_i^1 = C) = \theta_{i,1}$ (or $\pi_i^1(a_i^1 = D) = 1 - \theta_{i,1}$) and chooses second-order cooperation (or defection) with probability $\pi_i^2(a_i^2 = C) = \theta_{i,2}$ (or $\pi_i^2(a_i^2 = D) = 1 - \theta_{i,2}$). We also assume different agents choose actions independently: $\pi(a_1^1, a_1^2, \cdots, a_n^1, a_n^2) = \prod_{j=1}^n \pi_j(a_j^1, a_j^2)$. We can calculate the value function of agent $i$,

$$
\begin{aligned}
V_i^\pi =& \mathbb{E}_\pi[r_i(a_1^1, a_1^2, \cdots, a_n^1, a_n^2)] \\
=& \mathbb{E}_\pi\left[ \sum_{j=1}^n \frac{b}{n}\mathbb{I}(a_j^1 = C) - c\mathbb{I}(a_i^1 = C) - p\frac{1}{n}\mathbb{I}(a_i^1 = D)\sum_{j \neq i}\mathbb{I}(a_j^2 = C) \right. \\
& \left. - k\frac{1}{n}\mathbb{I}(a_i^2 = C)\sum_{j \neq i}\mathbb{I}(a_j^1 = D) \right] \\
=& \sum_{j=1}^n \frac{b}{n}\theta_{j,1} - c\theta_{i,1} - p\frac{1}{n}(1 - \theta_{i,1})\sum_{j \neq i}\theta_{j,2} - k\frac{1}{n}\theta_{i,2}\sum_{j \neq i}(1 - \theta_{j,1}) \\
=& (\frac{b}{n} - c)\theta_{i,1} + \sum_{j \neq i}\left( \frac{b}{n}\theta_{j,1} - p\frac{1}{n}(1 - \theta_{i,1})\theta_{j,2} - k\frac{1}{n}\theta_{i,2}(1 - \theta_{j,1}) \right).
\end{aligned}
$$

The learning process of agent $i$'s incentivizing policy can be framed as a bi-level optimization problem,

$$\max_{\pi_i^2 \in \Delta^2} V_i^{\pi'}$$

$$\text{subject to } \pi' = \prod_{j=1}^{n} \pi_j^{1,'} \pi_j^2$$

$$\pi_j^{1,'} = \arg\max_{\pi_j^1 \in \Delta^1} V_j^{\pi}, j = 1, \cdots, n$$

where $\Delta^1, \Delta^2$ is the valid space for environmental and incentivizing policies, respectively. The environmental policies of all the agents are first updated under the influence of incentivizing policies, and then the incentivizing policy of agent $i$ is optimized under the new environmental policies. It is challenging to get a closed-form solution for this optimization problem when other agents' incentivizing policies also optimize this bi-level optimization problem. But we can still provide insights into this problem. We first calculate the gradient of the value function $V_j^{\pi}$ w.r.t $\theta_{j,1}$:

$$\frac{\partial V_j^{\pi}}{\partial \theta_{j,1}} = \frac{b}{n} - c + p\frac{1}{n}\sum_{k \neq j} \theta_{k,2}. \tag{38}$$

With learning rate $\beta$, the updated $\theta'_{j,1}$ becomes:

$$\theta'_{j,1} = \theta_{j,1} + \beta \frac{\partial V_j^{\pi}}{\partial \theta_{j,1}} = \theta_{j,1} + \beta \left( \frac{b}{n} - c + p\frac{1}{n}\sum_{k \neq j} \theta_{k,2} \right). \tag{39}$$

Now the new value function for agent $i$ is

$$V_i^{\pi'} = (\frac{b}{n} - c)\theta'_{i,1} + \sum_{j \neq i} \left( \frac{b}{n}\theta'_{j,1} - p\frac{1}{n}(1 - \theta'_{i,1})\theta_{j,2} - k\frac{1}{n}\theta_{i,2}(1 - \theta'_{j,1}) \right). \tag{40}$$

To calculate updated incentivizing policies, we derive the gradient of $V_i^{\pi'}$ w.r.t. $\theta_{i,2}$. Note that agent $i$'s incentivizing policy does not directly influence its own environmental behaviors, so we have

$$\frac{\partial \theta'_{j,1}}{\partial \theta_{i,2}} = \begin{cases} 0, & j = i, \\ \frac{\beta p}{n}, & j \neq i. \end{cases}$$

With this derivative in hand, we can obtain the gradient of $V_i^{\pi'}$ w.r.t $\theta_{i,2}$:

$$\frac{\partial V_i^{\pi'}}{\partial \theta_{i,2}} = \sum_{j \neq i} \left( \frac{b}{n^2}\beta p - k\frac{1}{n}(1 - \theta'_{j,1}) + \beta p k \frac{1}{n^2}\theta_{i,2} \right)$$

$$= -k\frac{1}{n}\sum_{j \neq i}(1 - \theta_{j,1}) + \beta\frac{1}{n} \left[ (n-1)\left( \frac{b}{n}(p+k) - kc \right) + pk\frac{1}{n}\left( (n-2)\sum_{j=1}^{n}\theta_{j,2} + n\theta_{i,2} \right) \right].$$

We find that the learning rate $\beta$ has a decisive influence on the positivity and negativity of the above equation, and the direction of the gradient w.r.t $\theta_{i,2}$ cannot be determined at each step. An alternative is to consider multiple gradient steps, calculate optimal $V_i^{\pi}$ given incentivizing policies, and then derive the change direction of $\theta_{i,2}$ based on convergent $V_i^{\pi}$. However, it is challenging to formally calculate the global optimal $V_i^{\pi}$.

In addition, another problem arises when we consider the homophily constraint as a loss function. Using the definition of Eq. 4 and Eq. 5 in the paper, the homophily loss of agent $i$ under the updated environmental policies is

$$\mathcal{L}_i^{\pi'} = -\sum_{j \neq i, k \notin \{i,j\}} (1 - \theta'_{k,1})\left( \theta'_{i,1}\theta'_{j,1} + (1 - \theta'_{i,1})(1 - \theta'_{j,1}) \right)\left( \theta_{j,2}\log\theta_{i,2} + (1 - \theta_{j,2})\log(1 - \theta_{i,2}) \right). \tag{41}$$

With further calculation, we can find that the derivative of $\mathcal{L}_i^{\pi'}$ w.r.t $\theta_{i,2}$ has a nonlinear dependency on others' policies, which makes the dynamics of the system hard to capture, not to mention obtaining the closed-form solution (Perko, 2013).

So far, we have presented the challenges of obtaining theoretical supports in temporally extended cases. These challenges have perplexed the researchers for a long time (Gould et al., 2016; Hirsch et al., 2012; Hughes et al., 2018), and we believe that the solution for these questions is an important and promising direction for future work.

### F.2 LIMITATIONS OF EMPIRICAL IMPLEMENTATIONS

In this work, we encourage homophily on the level of incentivizing behaviors and outperform other works in SSDs. However, computing the homophily loss in Sec. 4 requires observing other agents' environmental actions, as well as their incentivizing actions towards all other agents. It might be a promising future work to restrict the agents' observaton and only allow the agent to incentivize nearby agents. Opponent modeling and communication might be critical techniques to achieve these goals.

In addition, incentivizing actions are expected to positively influence the return given by the environment through incentivizing other agents. However, incentivizing actions mainly influence the *policy update* of other agents, so the effects of incentivizing actions could be delayed, which may result in slow learning. Yang et al. (2020) have studied this issue on the framework of actor-critic, but how to effectively solve this problem on the framework of Q-learning is still an open question. Fortunately, one advantage of homophily is that its influence is immediate. If homophily can help cooperation emergence (and fortunately it can, as shown by our results), this timely feedback can bypass the problem of delayed reward influence. In the meantime, we are aware that addressing the problem of delayed reward influence may further improve learning performance. We will study this more in-depth in future work.

