# OpenReview forum: "Learning Homophilic Incentives in Sequential Social Dilemmas"
_ICLR.cc/2022/Conference — ICLR 2022 Submitted_

### Official Review · Reviewer_neVn · 2021-10-30

**Correctness:** 1
**Technical Novelty And Significance:** 4
**Empirical Novelty And Significance:** 4
**Recommendation:** 3
**Confidence:** 4

**Main Review:**

Strengths:
1. The writing is generally well-organized and of good quality. The detail introduction, related work and motivating example section served nicely to establish the current proposal’s novelty.
2. This paper, as far as I know, is the first to try to address second-order social dilemmas in MARL, which should be the main reason for the existing baselines requiring 100M to learn cooperative strategies.
3 This paper makes an adequate comparison with various baselines.
Weaknesses
1.	The selection of colors in figs 1-3 makes a lot of confusing, sometimes bule means the 2-nd order defecting, sometimes it means one trajectory.
2.	It is not clear how the authors introducing homophily into the motivating example. What do you mean by “we encourage their incentives to be the same by converting the minority of P and C to the majority with a probability of 0.2.”
3.	Could you show more in the motivating example? I mean you can show that the mechanism behind the other baselines, such as Inequity Aversion (Hughes et al., 2018), Social Influence (Jaques et al., 2019) cannot solve 2nd-SDs. It is incredible that the baselines typically need hundreds of millions of steps to learn cooperation, but your methods only needs 2-5 million steps!!! Is the proposed method just a trick, which does not allow the agents to adopt 2-nd order defecting. What if some agents use your method, while some use other baselines?
4.	The equation for S^env (i,j) should be given in the main paper, because this is the main idea of this paper.

**Summary Of The Paper:**

How cooperative behavior emergeing among self-interested agents was a question of longstanding interest. In recent years, this problem has been investigated in multi-agent RL situations. This paper considers the instability of the cooperation in MARL.
The paper first illustrates the phenomenon of oscillation-producing second-order social dilemmas within an EGT case study, and shows how a homophily mechanism can resolve them and produce convergence to a socially efficient set of roles. Then they propose a loss function called “homophily loss” that can resolve these second-order dilemmas in MARL. They experiment with this in two SSD's, Harvest and Cleanup, and show that their method outperforms the baselines they investigate.

**Summary Of The Review:**

The paper is generally well written, and with some novelty. But the provided results cannot convince me that the proposed method is not just a trick. I will rasie my points if the authors can provide more results to convince me.

---

> ### Author Response · Authors · 2021-11-23
> **Reply to Reviewer neVn**
>
> We thank the reviewer for the comments and suggestions. Here we provide detailed clarifications to your questions. If you have any further questions or comments, please post them and we will be happy to have further discussions.
>
> > The selection of colors in figs 1-3 makes a lot of confusing, sometimes blue means the 2nd order defecting, sometimes it means one trajectory
>
> Thanks for the suggestions, we will fix them in the next version.
>
>
> > It is not clear how the authors introduced homophily into the motivating example.
>
> To introduce homophily into the motivating example, we encourage agents with similar environmental behaviors to have similar incentivizing behaviors. In the motivating examples, only Contributors(C) and Punishers(P) have the same environmental behaviors, i.e. contributing to the public goods. Thus we only need to encourage their incentives to be the same. We realize this by converting the minority of Punishers and Contributors to the majority of them with a probability of $\lambda$ ($\lambda$ in the paper).
>
>
> > Could you show more in the motivating example? Such as other baselines (Inequity Aversion, Social Influence).
>
> Thanks for the suggestions, and we have shown more details in Appendix B.4 and Figure 12 in the updated paper. The cooperative area volume of inequity aversion is smaller than our method (about 60\%). Moreover, unlike our method, there is no cooperative area around the vertex N. Once the population of agents is trapped in the plane C-D-N, they can never escape from it.

---

### Official Review · Reviewer_bDVR · 2021-11-04

**Correctness:** 3
**Technical Novelty And Significance:** 2
**Empirical Novelty And Significance:** 2
**Recommendation:** 3
**Confidence:** 5

**Main Review:**


I think the extension of what has been studied purely in the PD to more complex games is an interesting idea. The experiments are well done.

However, I’m not sure there is enough in this paper to warrant publication at ICLR.

As the authors point out, the evolution of second order cooperation (i.e. punishing of non-punishers) is heavily studied in the standard evolution of cooperation literature.

As the authors point out, there is already work in the standard PD space that shows that preferential matching (which is quite similar to what the authors call homophilic matching) can stabilize cooperation in the PD. Note that the authors cite a few of the main papers in this literature but there are many more, as this topic has been done pretty extensively. I would actually remove section 3 from the paper and just state the headers as a sentence as these things have hundreds (thousands?) of pages already written about them.

Given that this is so well understood in the PD, it seems like the main novelty in this paper is just the extension of the PD to the SSD and the use of RL instead of e.g. Moran processes. I am not sure that’s a sufficient contribution.

Possible Extensions
I do not want to sound too negative, I think this is an interesting start, so I now discuss some potential extensions to increase the meat in the paper.

I would recommend the addition of theoretical guarantees (fine if they’re just in tabular RL). A paper of just experiments begs the question of how cherry picked the environments are and what the boundary conditions are. This would be a real extension since most of the existing literature only has theoretical results for the PD, which, as the authors point out, is very simple and unclear how much generalizes.

The homophily loss in the paper is basically just biasing the learning towards strategies which do well when paired with themselves. This actually smells a bit like the concept of evolutionary stability, so I wonder if that is a base for a potential theorem.

**Summary Of The Paper:**

Summary
The paper studies the evolution of cooperation in SSDs which are essentially more complex versions of the (repeated) Prisoner’s Dilemma. Whether cooperation in a population evolves (or not) depends on whether there exist individuals who punish non-cooperators. This evolution is now doubly fragile because there is an incentive to free-ride on punishment – that is, not to punish the punishers. The authors introduce a mechanism where agents also try to seek out those who have similar strategies as them, which reduces second order free riding.


**Summary Of The Review:**

 I’m not sure there is enough new stuff in this paper to warrant publication at ICLR.

---

> ### Author Response · Authors · 2021-11-23
> **Reply to Reviewer bDVR**
>
> Thank the reviewer for the comments and suggestions. Below we address the reviewer's concerns in detail, and we are open to any further discussions.
>
> > As the authors point out, the evolution of second order cooperation (i.e. punishing of non-punishers) is heavily studied in the standard evolution of cooperation literature.
>
> Previous works [1,2,3] that considered second-order social dilemmas focused on non-sequential social dilemmas (SDs). In contrast, our work presents a novel learning approach to addressing second-order social dilemmas in complex sequential social dilemmas (SSDs), which raise new challenges:
>
> (1) Cooperation is more vulnerable to first-order and second-order social dilemmas in SSDs than in SDs [4], and thus SSDs often require a different **computational** approach for cooperation emergence from that in SDs, which motivates our work.
>
> (2) It is common to assume to know whether a certain strategy is cooperative or defective in SDs [5,6], but this assumption does not naturally hold in SSDs. This is because temporally extended strategies are much more complex and can be dynamically mixed with cooperation and defection, especially during learning.
>
>
> > As the authors point out, there is already work in the standard PD space that shows that preferential matching (which is quite similar to what the authors call homophilic matching) can stabilize cooperation in the PD.
>
> Most prior works only proposed the concept of homophily in SDs (e.g., [7]) and homophily in SSDs was considered "*not a general method for multi-agent reinforcement learning, since honest signals of cooperativeness are not normally observable in the ISD models typically studied in deep reinforcement learning*" [8]. That is to say, although the concept of homophily is not new, how to effectively enable homophily in SSDs was an open problem. Our paper presents a novel learning framework that effectively computerizes the concept of homophily in complex sequential social dilemmas and significantly outperforms other SoTA algorithms on learning cooperation in SSDs.
>
>
> > The main novelty in this paper is just the extension of the PD to the SSD and the use of RL instead of e.g. Moran processes.
>
> This claim is clearly not well justified. Our work is novel and different from the papers mentioned in our paper in a multi-dimension perspective. For example, the differences are summarized as follows.
>
> **(1)** We show that second-order social dilemmas are key factors that can lead to unstable cooperation in SSDs, and policy learning is an extra factor exacerbating second-order social dilemmas. In contrast, other works only consider second-order social dilemmas in non-sequential SDs [1,2,3].
>
> **(2)** We encourage homophily on the level of incentivizing behaviors to promote cooperation, while other works only consider environmental assortative matchmaking [7,8]. However, encouraging homophily on environmental behaviors in SSDs is not as effective as it does in SDs.
>
>
> > I would recommend the addition of theoretical guarantees (fine if they're just in tabular RL)
>
> We thank the reviewer for the suggestions. Actually, we have tried to provide theoretical guarantees in Appendix F.1. However, we found that it presents unique challenges to obtain theoretical supports in even tabular RL. These challenges have perplexed the researchers for a long time, and we believe that providing theoretical guarantees for our method is an important and promising direction for future work. For more details, please refer to Appendix F.1.
>
> ---
>
> [1] Schmid, Kyrill, Lenz Belzner, and Claudia Linnhoff-Popien. "Learning to Penalize Other Learning Agents." *ALIFE 2021: The 2021 Conference on Artificial Life*. MIT Press, 2021.
>
> [2] Rand, David G., and Martin A. Nowak. "The evolution of antisocial punishment in optional public goods games." *Nature communications* 2.1 (2011): 1-7.
>
> [3] Úbeda, Francisco, and Edgar A. Duéñez‐Guzmán. "Power and corruption." *Evolution: International Journal of Organic Evolution* 65.4 (2011): 1127-1139.
>
> [4] Leibo, Joel Z., et al. "Multi-agent reinforcement learning in sequential social dilemmas." arXiv preprint arXiv:1702.03037 (2017).
>
> [5] Anastassacos, Nicolas, Stephen Hailes, and Mirco Musolesi. "Partner selection for the emergence of cooperation in multi-agent systems using reinforcement learning." Proceedings of the AAAI Conference on Artificial Intelligence. Vol. 34. No. 05. 2020.
>
> [6] Hauert, Christoph, et al. "Via freedom to coercion: the emergence of costly punishment." science 316.5833 (2007): 1905-1907.
>
> [7] Fletcher, Jeffrey A., and Michael Doebeli. "A simple and general explanation for the evolution of altruism." *Proceedings of the Royal Society B: Biological Sciences* 276.1654 (2009): 13-19.
>
> [8] Wang, Jane X., et al. "Evolving intrinsic motivations for altruistic behavior." *arXiv preprint arXiv:1811.05931* (2018).

---

### Official Review · Reviewer_TQpe · 2021-11-05

**Correctness:** 3
**Technical Novelty And Significance:** 2
**Empirical Novelty And Significance:** 3
**Recommendation:** 5
**Confidence:** 3

**Main Review:**

On the positive side, I found the paper to be well-written, and it gives a fairly satisfying and complete first investigation of using the tendency of homophily in sequential social dilemmas. The idea behind homophily is intuitive, i.e. associate human individuals with similar others, and seems to be quite useful according to the simulation results in both single-stage and sequential social dilemmas.

On the other hand, a major concern in my view is that the computation of S^{env} and S^{inc} in equation (4) assumes the agent knows all the other agents’ incentivizing and environmental information. This doesn’t seem to be super realistic in practice given that all the agents are self-interested. In addition, the proposed method requires much more information about other agents, so it’s not very surprising that the proposed method can achieve better cooperation between agents compared to other baseline methods, which uses much less information about other agents.  The sharing of this environmental and incentivizing information between all the agents can already be seen as a kind of “cooperation” imposed by the proposed method. Therefore, it is not truly clear to me that the better performance of the proposed method is due to more demanding requests of the agent, or due to a better learning algorithm. This is a major reason for my rating of weak rejection.

In addition, how to specifically implement the homophily, e.g. to what extent (especially this paper is using a binary instead of a continuous variable to determine the similarity in the sequential social dilemmas setting) can we regard two individuals as similar? In Section 3.D, the authors introduced a concept - degree of homophily, and show the “safer region” increases with the degree of homophily. How is this degree estimated, does this degree exist in practice, and if so what’s a normal value for this? Also, what’s the relationship between the degree and the S^{env} (environment similarity variable) in the sequential social dilemmas setting? There seems not a good guidance for these questions.

Additionally, the title of the paper emphasizes “sequential social dilemmas”. However, I am not sure if this matches the main concern of the paper. As far as I understand, the main contribution of the paper is using homophily to solve the second-order social dilemmas resulting from the incentive mechanisms other than temporally extended mechanisms. What’s more, the formal analysis of how homophily solves second-order social dilemmas is only proved for single-stage social dilemmas, only simulation results were provided for this paper’s main goal - sequential social dilemmas. Therefore, I think the results from this paper don't solve the problem as completely as proposed.

About the simulation results, I like the trajectory figure and gradient field figure plotted by the authors. However, the results there need more explanation. Why does oscillating cooperation mean the distribution of population members converges to Nonparticipants (N)? I think when the distribution of population members converges to Defecting agents, this can also be thought as noncooperation. Similarly, why does the convergence of the distribution to Punishers (P) means stable cooperation?

Some notation issues:
Section 3 Paragraph 1: no definition about 1st-SDs, is it just the SDs? Or if there are others that the authors have in mind;
Section 2 Paragraph 1: SDs is defined as the abbreviation of Social dilemmas which is contradicting with the definition of Non-sequential
Social Dilemmas from Section 1 Paragraph 2.


**Summary Of The Paper:**

This paper studies the problem of promoting cooperation among self-interested agents. Inspired by the work which studied homophily in non-sequential social dilemmas, the authors proposed to use homophily, which encourages agents with similar environment behaviors to have similar incentivizing behaviors, to achieve stable cooperation in sequential social dilemmas. Simulation evidence from sequential social dilemmas of public goods and tragedy of the commons was also presented in the paper to support the effectiveness of the proposed method.

**Summary Of The Review:**

 Overall, I do think that, unless there is something I am missing, the limitations I described somewhat limit what we can take away for realistic settings, and so doesn’t quite settle the questions claimed so completely as suggested. I would recommend weak
rejection.

---

> ### Author Response · Authors · 2021-11-23
> **Reply to Reviewer TQpe**
>
> We thank the reviewer for the insightful comments and suggestions. We provide further clarifications as below and hope they address your concerns. If you have any other questions, we will be happy to have further discussions.
>
> > The computation of $S^{env}$ and $S^{inc}$ in equation (4) assumes the agents knows all the other agents' incentivizing and environmental information.
>
> We re-evaluate our method when the incentive range is restricted to the view size in Appendix B.5 and Figure 13 in the updated paper. Now the agents can only incentivize other agents in the view range. We found that there is only a slight decrease in the performance compared to the unrestricted version, which shows the robustness of our method.
>
>
> > The proposed method requires much more information about other agents.
>
> Not really. We speculate that the reviewer is referring to the behavior features we used to measure the environmental behavior similarity. However, these features are also used in other baselines. Concretely, the features include the changes made to common information like the amount of harvesting, the amount of cleanup. The amount of harvesting is actually the environment rewards, which are used in Inequity Aversion baseline. And the amount of cleanup can be calculated from agents' observations.
>
>
> > This paper is using a binary instead of a continuous variable to determine the similarity in sequential social dilemmas.
>
> The use of binary variables to determine the similarity in sequential social dilemmas is mainly for stability and simplicity.
>
> (1) Stability. To use a continuous variable to determine the similarity, we may need to learn a parametric measuring function. However, we may get trivial solutions as we discussed in Sec.4.2 in the paper. Instead, we use a simple non-parametric clustering method --  X-means.
>
> (2) Simplicity. We are not determining whether two policies are exactly the same, but whether they are similar over a period of time (e.g. 10 steps in our experiments). The diversity of the policy in a short period is much less than the policy over an episode. Thus it is suitable and simple to just use a binary variable.
>
>
> > Questions regarding the concept -- "degree of homophily".  How is this degree estimated, does this degree exist in practice, and if so what’s a normal value for this? Also, what’s the relationship between the degree and the $S^{env}$ (environment similarity variable) in the sequential social dilemmas setting?
>
> The degree of homophily is a predefined hyper-parameter. In the motivating example, the degree is defined as the probability $\lambda$ ($\lambda$=0.2 in the paper) of converting the minority of punishers and contributors to the majority of them. Please refer to Appendix B.2 for more investigations of this degree. In sequential social dilemmas, the degree is defined as the loss scaling factor of homophily loss $\lambda^{\text{homo}}$ ($\lambda^{\text{homo}}$=0.01 in the paper). In both cases, this degree of homophily can be interpreted as the strength of homophily tendency.
>
>
> > Why does oscillating cooperation mean the distribution of population members converges to Nonparticipants (N)?
>
> The population does not converge to Nonparticipants (N), but only rotates counterclockwise in the vicinity of vertex N (Sec.3 Part A).
>
>
> > I think when the distribution of population members converges to Defecting agents, this can also be thought as noncooperation.
>
> Defecting agents can be thought as noncooperation, but not non-participation. As we defined in Sec.3, the Defectors do not contribute to but benefit from the public goods, and the Nonparticipants neither contribute to nor benefit from the public goods.
>
>
> > Why does the convergence of the distribution to Punishers (P) means stable cooperation?
>
> The Punishers can be viewed as cooperators, because they also contribute to the public goods as the Contributors.
>
>
> > Some notation issues.
>
> Thanks for pointing them out, and we have fixed them in the updated paper.

---

### Author Response · Authors · 2021-11-23
**General Response: Summary of Paper Updates**

We thank all reviewers for their reviews and suggestions of our work. All the comments are helpful and have been reflected in the updated paper. Here we briefly list the updates.

**[Appendix B.4]** To show the mechanisms behind other baselines, we plot the gradient field of Inequity Aversion.

**[Appendix B.5]** We restrict the incentive range of each agent, so that the agents can only incentivize other agents in the view range. The learning curve shows the robustness of our method.

---

### Decision · Program_Chairs · 2022-01-20

**Decision:**

Reject

**Comment:**

This paper propose a novel framework to increase cooperation in second-order social dilemmas. This is based on encouraging homophilic incentives. Reviewers agree that the paper does not meet the standards of publication yet. In particular, they worry that the assumptions made are so restrictive as to make model inapplicable to interesting problems. There is also a concern that the work is simply not novel enough.